# HOTRUNZ: an open access 1-km resolution monthly 1910-2019 time-series of interpolated temperature and rainfall grids with associated uncertainty for New Zealand

Thomas R. Etherington[1], George L. W. Perry[2], Janet M. Wilmshurst[1]

[1]Manaaki Whenua – Landcare Research, Lincoln 7608, New Zealand
[2]School of Environment, University of Auckland, Private Bag 92019, Auckland, New Zealand

*Correspondence to*: Thomas R. Etherington (etheringtont@landcareresearch.co.nz)

**Abstract.** Long time-series of temperature and rainfall grids are fundamental to understanding how these environmental variables affect environmental or ecological patterns and processes such as plant distributions, plant and animal phenology, wildfires, and hydrology. Ideally such temperature and rainfall grids are openly available and associated with uncertainties so that data-quality issues are transparent to users. We present a History of Open Temperature and Rainfall with Uncertainty in New Zealand (HOTRUNZ) that uses climatological aided natural neighbour interpolation to provide monthly 1-km resolution grids of total rainfall, mean air temperature, mean daily maximum air temperature, and mean daily minimum air temperature across New Zealand from 1910 to 2019. HOTRUNZ matches the best available temporal extent and spatial resolution of any open access temperature and rainfall grids that include New Zealand, and is unique in providing associated spatial uncertainty in the variables' units. The HOTRUWNZ grids capture the dynamic spatial and temporal nature of monthly temperature and rainfall and the uncertainties associated with the interpolation. We also demonstrate how to quantify and visualise temporal trends across New Zealand that recognise the temporal and spatial variation of uncertainties in the HOTRUNZ data. The HOTRUNZ data is openly available at  https://doi.org/10.7931/zmvz-xf30 (Etherington et al., 2021).

## 1 Introduction

Climatologies such as WorldClim 2 (Fick and Hijmans, 2017) and CHELSA (Karger et al., 2017) that provide spatial grids (or: layers, surfaces) consisting of cells (or: grid points, pixels) that record climatic variables such as temperature and rainfall underpin thousands of environmental and ecological studies. These climatologies represent long-term $\approx$ 30-year climate averages, but long time-series of temperature and rainfall grids each of which covers a shorter period are also highly desirable as they can be correlated with long-term environmental or ecological data to explore how temperature and rainfall influence such processes. For example, monthly time-series data of temperature and rainfall conditions can be used to improve the understanding of plant distributions (Stewart et al., 2021), plant and animal phenology (Gordo and Sanz, 2005), wildfire histories (Girardin and Wotton, 2009), and hydrology (Remesan et al., 2014).

Meteorologists and climatologists are recognising the importance of implementing open access methodologies, as openly sharing data, source code, and knowledge provides exciting opportunities for scientific discovery (de Vos et al., 2020). For New Zealand there are limited time-series options for national-scale temperature and rainfall grids (Table 1), and while each varies in its spatial and temporal characteristics, no one open access dataset has the combined criteria of $\leq 1$ km$^2$ spatial and

monthly temporal resolution over the last century. Producing such historical temperature and rainfall data is challenging because weather station data are often sparse, especially in remote areas and for earlier time periods. However, this challenge should not preclude the creation of historical data, if we accept that we cannot be consistently successful for all locations and dates and we prioritise providing temperature and rainfall data with associated uncertainty data.

Much emphasis has been placed on quantifying (Foley, 2010) and visualising (Retchless and Brewer, 2016) the uncertainty of future climates. Quantifying and visualising uncertainty is also critical for the judicious use of historic climate and weather data, as even when following open science practices that involve sharing data and code it can still be difficult to ensure that an end-user is aware of and understands the quality of the data (de Vos et al., 2020). Historical climate and weather data are often used predictively, but these predictions must recognise the underlying uncertainties so that their reliability can be understood

and communicated. The uncertainty (or reliability) of weather and climate spatial data is often quantified with metrics such as the root-mean-square error (RMSE) or the mean absolute error (MAE) derived from cross-validation that iteratively excludes each weather station and then interpolates a value for the weather station using the remaining data (Willmott and Matsuura, 2006). However, these are global metrics that describe uncertainty with a single spatially averaged value and provide no information about how the uncertainty will vary across space (Zhang and Goodchild, 2002). When estimates of

spatial uncertainty accompany climate and weather data, they typically provide only simple indications of areas that are reliable (Abatzoglou et al., 2018;Harris et al., 2020b), which limits how uncertainty can be incorporated into analyses as estimates from individual cells are not matched with an uncertainty in the variables' units.

    To facilitate understanding of how long-term temperature and rainfall patterns have been changing and potentially affecting

environmental and ecological processes in New Zealand, we use climatologically aided interpolation to produce a History of Open Temperature and Rainfall with Uncertainty in New Zealand (HOTRUNZ). HOTRUNZ is an openly available history of monthly temperature and rainfall data for 1-km$^2$ grid cells across New Zealand that matches the best available spatial and temporal extent and resolution of any currently available open data and is unique in providing associated spatial uncertainty grids in each variables' units (Table 1).

## 2. Weather and climatology data

Precomputed monthly statistics calculated from weather station data are freely available from New Zealand's National Climate Database (NIWA, 2020). For New Zealand's three main islands and their associated near-shore islands we queried the database for data from 1900 to 2019 for monthly statistics of: total rainfall, mean air temperature, mean daily maximum air temperature, and mean daily minimum air temperature. Using older weather station data can be problematic as the manual nature of the recording can lead to gaps in the records over time and locations can be poorly recorded. Therefore, to ensure a high quality of weather station data we only used monthly statistics data for weather stations that were recorded in the database as having a complete set of daily records for every day of the month and hence did not contain any estimated values. We also only used data from weather stations whose locations were recorded in the database as being reliable to within 100–200 m at worst and therefore could be reliably located within a 1-km$^2$ grid cell.

Having applied these filters to the weather station database we had data from 3438 weather stations across New Zealand, but most of these stations were from lower elevational areas, with 89 % below 500 m (Figure 1). This lack of weather station data at higher elevations has been noted before, and is in part why interpolations of temperature and rainfall data in New Zealand are usually less accurate at higher elevations (Tait and Macara, 2014;Tait et al., 2012). To recognise the challenges of interpolating temperature and rainfall at higher elevations we will follow Tait et al. (2012) and will evaluate the accuracy of our data for locations below and above 500 m elevation (Figure 1). Further complexity is added to data availability in that many of the weather stations were short-lived or provided intermittent records, so the amount of data available in any given month varied through time, across space, and between variables, but there were some consistencies. There was much more rainfall data than temperature data, there was more data in recent times (with a peak around 1980), and there was always less data in the mountainous interiors of both islands and in the more remote southerly and westerly regions of New Zealand (Figure 2).

Climatologies that describe average weather patterns over several decades have underpinned successful previous rainfall interpolations in New Zealand (Tait et al., 2006). Therefore, we used a climatologically aided interpolation approach by interpolating monthly values as an anomaly from a climatic normal (Willmott and Robeson, 1995) as this technique has been successfully applied elsewhere (Abatzoglou et al., 2018;Harris et al., 2020b;Hofstra et al., 2008). The basic premise of climatologically aided interpolation is that rather than directly interpolating weather station data in each month, by using an underlying long-term climatology grid for that month, monthly anomalies can be calculated as the difference between the weather station value and the climatology grid value. These anomalies are then interpolated and added to the climatology grid to estimate how the conditions in that month deviated from the climatological normal. When compared to the same interpolation method using just weather station data climatologically aided interpolation is advantageous in that it: (i) has reduced errors, (ii) has errors that are insensitive to changes in the number of weather stations, and (iii) can indirectly account

for topographic effects when climatologies with high spatial resolution are used (Willmott and Robeson, 1995). Openly available New Zealand climatology grids for the rainfall and temperature variables giving the average conditions over the thirty year period from 1950-1980 for each month at 100 m grid cell resolution (McCarthy et al., 2021;Leathwick et al., 2002) were used to produce 1-km$^2$ grid cell resolution climatologies as the basis of the climatologically aided interpolation. These climatologies were developed from long-term weather station data interpolated using thin-plate splines using geographic variables of elevation, and in the case of rainfall an east-west topographic protection variable (Leathwick et al., 2002). Using these climatologies for our climatologically aided interpolations incorporates these geographic variables indirectly into our interpolations, and by having a climatology for each month, as opposed to a single climatology for a whole year, seasonal shifts in temperature and rainfall are also accounted for.

## 3. Interpolation with uncertainty

We selected natural neighbour (or Sibson) interpolation (Sibson, 1981) to interpolate the anomalies. Natural neighbour interpolation has been shown to perform well for interpolating rainfall and temperature data (Hofstra et al., 2008;Keller et al., 2015;Lyra et al., 2018). More specifically we chose to use natural neighbour interpolation because it is: (i) an *exact interpolator*, meaning it will retain the original data values at locations with input data in the interpolated grid and will only interpolate within the range of the original data and so cannot produce wildly unrealistic interpolations; (ii) a *local method*, which interpolates for a location only using data from that location's immediate surrounds; (iii) *spatially adaptive*, so automatically adapting to localised data distribution and density; (iv) not based on fitting statistical trends and so does *not require large sample sizes* (Etherington, 2020). These properties are desirable given our interpolations will need to adapt to the increasingly sparse and irregular data that occurs further back in New Zealand history, and which preclude using more complex interpolation methods that require more data. For example, when interpolating at higher elevations where there are fewer data, simpler methods can perform better than more complex methods (Stahl et al., 2006). Therefore, in choosing natural neighbour interpolation we do not suggest that it is universally the best interpolation method, but rather it was the most appropriate given our data situation and interpolation objectives.

We applied a discrete (or digital) form of natural neighbour interpolation (Park et al., 2006) that simultaneously calculates uncertainty as a cross-validation error-distance field (Etherington, 2020). This interpolation method works by defining a grid of cells for which interpolated values will be calculated. Data cells are first defined as those cells that contain weather station data. Where there are multiple weather stations within a single data cell the data cell is given the mean value of any weather stations they contain, therefore with discrete natural neighbour interpolation it is not the number of weather stations but rather the number of data cells that determines the amount of data available for the interpolation. The number of data cells and the spatial pattern of the data cells, as measured by mean nearest neighbour distance (Clark and Evans, 1954), varied over time

(Figure 3). At elevations < 500 m the number of data cells for all variables increases steadily over time with a peak around the 1980s followed by a decline that while continual for rainfall is then reversed for all temperature variables. While the number of data cells varies, the mean nearest neighbour distance is reasonably constant over time indicating that spatial coverage by the weather station network has been consistent despite variations in the number of data locations. Spatial coverage does decrease moving back through time, and that coverage was particularly poor for the temperature variables pre-1910. At elevations ≥ 500 m there are far fewer data cells at all points in time, and pre-1910 there were usually no data cells available for the temperature variables. However, the mean nearest neighbour distance of the data cells ≥ 500 m is similar to the data cells < 500 m. This difference in number of data cells but consistency in spatial pattern between the elevational regions suggests that data cells ≥ 500 m tend to occur either along the edge of the ≥ 500 m region or are clustered within the ≥ 500 m region, which is consistent with the general pattern of all weather stations providing data (Figure 1). Overall, these patterns in data cell numbers and spatial pattern indicate a need for caution in our interpolations in regions ≥ 500 m elevation and pre-1910.

Once the data cells for each month are established, the discrete natural neighbour interpolation proceeds by assigning all other cells the value of the nearest data cell (Figure 4a). The interpolated value for a grid cell is the mean of the grid cell values that are as close or closer to the interpolation cell than a data cell (Figure 4b), which, when repeated across all grid cells, yields a smooth interpolation grid (Figure 4c). A cross-validation error-distance field (Etherington, 2020) is then used to quantify the interpolation uncertainty. In essence uncertainty increases with distance from a data cell, with zero uncertainty at the data cells themselves where the underlying value is known and does not require interpolation. The rate at which uncertainty increases with distance from data cells is based on a cross-validation of the data cells that calculates the absolute interpolation error and distance to other data cells. This approach ultimately results in areas having greater uncertainty when they are more distant from data cells and are near data cells that would be harder to interpolate accurately when absent. Uncertainty is calculated using the discrete natural neighbour interpolation process to interpolate the distances to the data cells to produce natural neighbour distances (Figure 4d) and to interpolate the cross-validated error rate, as the ratio of the cross-validated absolute error to natural neighbour distance, for each data cell (Figure 4e). The product of the natural neighbour distances and cross-validated error rates produces a cross-validation error-distance field (Figure 4f) that yields a grid of interpolation uncertainties in the same units as the interpolation and is highest in areas that are more distant from data cells and in areas where the variable being interpolated has higher spatial heterogeneity and therefore is harder to interpolate accurately when data is sparse (Etherington, 2020).

**The resulting temperature and rainfall grids appeared to capture the heterogeneous and dynamic nature of the monthly temperature and rainfall variables and the uncertainty associated with the interpolation. For example, the May total rainfalls and uncertainties (Figure 3) illustrate the dynamic nature of the monthly rainfall and the associated**

uncertainty. **There are clear differences in total rainfall in May over time and shifts in the location of the wettest regions. Our emphasis on quantifying uncertainty is justified by the magnitude and location of uncertainty changing**
**through time. This emphasises our view that global error estimates for large areas and timeframes are potentially unhelpful, as the degree of uncertainty can change rapidly over space and time, and so all interpolation estimates should be associated with an individual matching uncertainty estimate. We interpret this rapidly changing uncertainty to result primarily from the data limitations of interpolation, as higher uncertainty occurs in locations more distant from temperature or rainfall data whose location and abundance change over space and time (Figures 2 and 3). However,**
**uncertainty can be high in regions where temperature and rainfall data are available, which we interpret as arising from the spatial variability of individual monthly temperature and rainfall patterns, as when spatial variability of temperature and rainfall patterns increases over shorter distances, such as in mountainous terrain, interpolation becomes increasingly uncertain (Etherington, 2020).4. Interpolation evaluation**

The MAE provides a reasonable estimate of the actual error rates for discrete natural neighbour interpolation (Etherington,
2020). Therefore, even though each monthly temperature and rainfall grid has a matching uncertainty grid, we also used cross-validation to calculate the MAE for each monthly interpolation to validate the method and facilitate comparisons with other temperature and rainfall interpolations (Willmott and Matsuura, 2006).

While the number of data cells (which equates closely to the number of weather stations) used during interpolation varied over
time, the distance between data cells remained reasonably constant (Figure 3). This consistency of spacing between data cells may explain why the MAEs associated with all the temperature and rainfall variables remained reasonably constant around 18 mm for total rainfall < 500 m and 24 mm for rainfall $\geq$ 500 m, and around 0.5 $^{\circ}$C for all temperature variables < 500 m and 0.7 $^{\circ}$C for all temperature variables $\geq$ 500 m (Figure 6). As might be expected, MAEs were lowest when interpolating data during 1950-1980 which is the period the climatologies aiding the interpolation were created for, and at lower elevations for
which there is more data (Figure 3). There was also an increase in MAE for some variables moving away from the 1950-1980 climatology period that becomes more pronounced pre-1910 as weather station data availability becomes extremely limited resulting in as few as 53 to 104 rainfall and 3 to 17 temperature data cells across New Zealand in each month, with no data cells for temperature variables $\geq$ 500 m pre-1910,. We conclude that the data are most reliable from 1910 to 2019, representing the four decades either side of the 1950-1980 climatologies used in the interpolations. Similar patterns to the MAE are seen
using the RMSE (Figure S1) and actual and estimated values are strongly positively correlated (Figure S2); thus, while these metrics are sensitive to outliers, we include them as additional information given their use in other temperature and rainfall interpolation evaluations (Hofstra et al., 2008).

We used the same cross-validation MAE approach to evaluate how well the estimated uncertainty matched the cross-validated
absolute errors (Figure 7). Again, performance was best during the 1950-1980 climatology period, with some variables showing more pronounced decreases in performance moving away from this time-period. The uncertainty error was around 28 mm for total rainfall < 500 m and 35 mm for rainfall $\geq$ 500 m, but there was a less pronounced difference between different elevations for temperature with all temperature variables at both elevations having an error of around 1 $^{\circ}$C. Similar patterns

to the MAE are seen using the RMSE (Figure S3) with a positive relationship between actual errors and estimated uncertainties
(Figure S4).

## 5. Using the uncertainty data

When using interpolated temperature and rainfall data users need to be able to make decisions specific to their requirements
(de Vos et al., 2020). Therefore, in HOTRUNZ we have endeavoured to match every interpolation with an individual measure
of uncertainty in the relevant units. However, we recognise that it is unusual for historical temperature and rainfall data to be
presented alongside such detailed uncertainty estimates. These uncertainty estimates provide a new and possibly challenging
analytical opportunity therefore we demonstrate how potential data users to incorporate the uncertainty data into an analytical
workflow and visualise the results.


A simple example of temperature and rainfall time-series analysis would be to use a Spearman's rank ($r_s$) correlation (Gregory,
1978;Spearman, 1904) to detect the directionality of any trends in temperature and rainfall patterns over time (Girardin and
Wotton, 2009). To incorporate uncertainty into this process, we adopt a Monte Carlo approach in which we produce many
equally possible temperature and rainfall histories by randomly sampling each month's temperature and rainfall as a random
value from a probability distribution. For our example analysis we simply use a uniform distribution with a range equal to the
interpolation uncertainty (limiting rainfall to a minimum of 0 mm). Many trends can then be calculated, with their distribution
used to infer the reliability of the analysis. For a location with high uncertainty the possible temperature and rainfall histories
can vary widely around the single interpolated temperature and rainfall history resulting in a wide distribution of possible
trends (Figure 8a). In other instances, the trend may be stronger than the uncertainty meaning that while the strength of the
trend varies its direction can be clearly established (Figure 8b). At locations where there is little uncertainty and hence minimal
variation in possible temperature and rainfall histories the trend can be established precisely (Figure 8c).

If this trend analysis with uncertainty process is repeated for every location, spatial trends can be analysed. The challenge is
then how to visualise the spatial pattern of uncertainty. Based on guidance relating to the cartographic visualisation of
uncertainty (Kaye et al., 2012;Retchless and Brewer, 2016), we selected a diverging colour scheme to show trends as the
median $r_s$ of the possible temperature and rainfall histories. To mask locations of increasing uncertainty we used a value-by-
alpha approach (Roth et al., 2010) that overlays a black mask that is increasingly opaque in locations of increasing uncertainty
that is measured as the 5[th] to 95[th] percentile range of the $r_s$ of the possible temperature and rainfall histories. The resulting map
indicates that in some regions there are clear trends in temperature and rainfall patterns over time, but in other regions the
uncertainty is too large to reliably make inferences from them (Figure 9). Areas of high uncertainty are not randomly

distributed, and are instead concentrated in areas with higher elevations and those with lower or more distant data availability (Figure 1). However, the exceptions to these general patterns again highlight the importance of providing specific uncertainty data for all interpolations. So, while our evaluation indicates that areas $\geq 500$ m elevation are generally less accurate than areas $< 500$ m elevation (Figure 6), users of HOTRUNZ should refer to the individual uncertainty data associated with their specific locations of interest because general rules may not always apply.

The results from our trend analysis (Figure 9) clearly demonstrate that it is possible to produce a long-term history of interpolated temperature and rainfall and emphasise the importance of quantifying the uncertainty of all interpolations. Of course, the approach we have used here is simply trying to illustrate the potential of the uncertainty data and should not be interpreted as the best or only analytical approach. It is obviously impossible for us to give guidance on how to incorporate uncertainty into all possible applications, but the Monte Carlo approach we present could be adapted to many situations. For example, the trend analysis shown here could be extended for those users interested in comparing long-term environmental data to weather data to identify associations between an environmental process and temperature and rainfall (Girardin and Wotton, 2009).

## 6. Limitations and future recommendations

We have stressed the importance of quantifying uncertainty when interpolating rainfall and temperature variables and have applied a novel approach that matches every interpolation estimate with an associated measure of uncertainty in the variables' units. However, while uncertainty in geographical information results from a combination of geographical abstraction, data acquisition, and geoprocessing (Zhang and Goodchild, 2002) our measure of uncertainty only captures the uncertainty associated with geoprocessing. Therefore, while the quantified uncertainty should alert potential users of locations where the interpolation is likely to be less reliable, potential users will still need to apply their own assessment regarding uncertainty associated with geographical abstraction and data acquisition. For example, we have used a geographical abstraction based upon 1-km$^2$ grid cells, but for some applications this abstraction may be too coarse, creating uncertainty about patterns at finer scales. Similarly, while we deliberately excluded unreliable weather station data, the amount of data varies through space and time, and uncertainty associated with interpolations will generally be higher in locations where there is less data and the spatial pattern of the variable being interpolated is more complex (Etherington, 2020).

While temperature and rainfall are key environmental variables influencing many ecological and physical processes, other variables could similarly explored for different contexts. Climatologies also exist for solar radiation, humidity, pressure deficit, and wind speed (McCarthy et al., 2021) for which matching monthly statistics from weather station data are available (NIWA,

2020); expanding the variable coverage to include these variables could be a useful addition to any future version of the data set.

Some aspects of the temporal and spatial scales of HOTRUNZ could be improved in subsequent refinements of this dataset. One limitation is that the monthly temporal resolution does not capture extreme but short duration weather events. For example, in July of 1996 there was an extreme cold snap that had significant effects on vegetation in southern New Zealand (Bannister, 2003) but is not evident in our data that averages minimum temperatures over the whole month. We only had access to monthly climatologies on which to base our interpolations, but if openly available weekly, or even daily, 
climatologies were created then it would be possible to interpolate historical weather at a finer temporal resolution to better capture extreme weather events of short durations.

While improving temporal resolution would require the creation of new climatologies, the spatial resolution could be improved up to 100 m with the climatologies used here. This improvement could be beneficial in the mountainous areas of New Zealand 
where temperatures can vary considerably within the 1-km resolution of our grids. Future improvements in temporal and spatial resolution would benefit from a more efficient computational workflow. Our computational workflow limited our processing to a 1-km resolution, but future versions could either leverage high performance computing, or to maintain a high degree of openness could continue to use desktop computing but with discrete natural neighbour interpolation leveraging the power of graphics processing units that are well suited to this method (Park et al., 2006).


We believe there is little point in extending the temporal extent of the data set. The growth of MAE associated with the reduction of temperature and rainfall data pre-1910 (Figure 4) indicates the available weather station data is insufficient to reliably estimate historical temperature and rainfall using our approach before the 20[th] century. While there are sources of additional temperature and rainfall data held in archives (Lorrey and Chappell, 2016) palaeo-environmental techniques may 
provide a better option for longer-term temperature and rainfall information (Cook et al., 2006;Duncan et al., 2010). There was a subtle increase in MAE towards the present that may be a function of a reduction in weather station data, particularly rainfall data, and perhaps a growing temporal mismatch between the 1950-1980 climatologies used to aid the interpolation (Figure 6). This indicates that continuing to use interpolation methods may become less feasible in the future. More recent climatologies could be produced to provide more temporally relevant climate data, but satellite data may provide a more useful 
data source for more recent temperature and rainfall history (Funk et al., 2015). We believe our interpolation approach is most valid for the pre-satellite era, and that by producing data that span part of the satellite era we provide a useful overlap for comparative purposes that could allow for a transition between historical temperature and rainfall data sources.

## 7. Data availability

The resulting HOTRUNZ data (Etherington et al., 2021) are openly available in non-proprietary file formats under a Creative Commons by Attribution 4.0 Licence and are archived at the Manaaki Whenua – Landcare Research DataStore https://doi.org/10.7931/zmvz-xf30.

## 8. Code availability

All HOTRUNZ analyses were conducted using an open-source software Python computational framework (Pérez et al., 2011) in conjunction with the gdal (GDAL/OGR contributors, 2021), pyproj (Snow et al., 2020), NumPy (Harris et al., 2020a), SciPy (Virtanen et al., 2020), numba (Lam et al., 2015), and Matplotlib (Hunter, 2007) packages. All the resulting code used to process data and plot figures presented here (Etherington, 2021) is openly available under an MIT Licence from the Manaaki Whenua – Landcare Research DataStore https://doi.org/10.7931/yk7g-vz81.

## 9. Conclusions

We present HOTRUNZ as *a,* rather than *the,* history of New Zealand temperature and rainfall; it would be possible to repeat the process using equally defensible quantitative methods and obtain different results. Likewise, changes in the spatial or temporal resolution will result in different patterns; however, there is no single *best* resolution as this will vary depending on the desired application. Nevertheless, as HOTRUNZ matches the highest available spatial and temporal extent and resolution of any currently available open access grids (Table 1), we believe that in creating HOTRUNZ we have significantly improved the ability for environmental and ecological scientists in New Zealand to understand how changing temperature and rainfall patterns have affected various environmental and ecological processes. Even with the spatially and temporally complex patterns of uncertainty, that are sometimes large, it is still possible to find consistent trends in temperature and rainfall (Figure 9). We hope our efforts to produce interpolation estimates with associated uncertainty, and examples of how to build that uncertainty into any analyses, will encourage the quantification and visualisation of uncertainty in weather and climate data sets elsewhere.

## Author contributions

TRE, GLWP, and JMW conceived and developed the idea. TRE developed the code and performed the data processing. TRE, GLWP, and JMW wrote the manuscript.

## Competing interests

The authors declare that they have no conflict of interest.


## Acknowledgements

This work was supported by: the Strategic Science Investment Funding for Crown Research Institutes from the New Zealand Ministry of Business, Innovation and Employment's Science and Innovation Group; and The University of Auckland Faculty Research Development Fund 3702237.

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

**Table 1: Datasets providing a time-series of spatial grids that estimate rainfall and temperature that include New Zealand.**

| Dataset | Licensing | Temporal extent | Temporal resolution | Spatial extent | Spatial resolution | Uncertainty grid |
|---------|-----------|-----------------|---------------------|----------------|---------------------|------------------|
| VCSN (Tait et al., 2012) | Proprietary | 1960-2022 | Daily | New Zealand | ≈ 5 km | No |
| WorldClim 2 (Fick and Hijmans, 2017) | Open | 1960-2018 | Monthly | Global | ≈ 4 km | No |
| CHELSA (Karger et al., 2017) | Open | 1979-2013 | Monthly | Global | ≈ 1 km | No |
| TerraClimate (Abatzoglou et al., 2018) | Open | 1958-2015 | Monthly | Global | ≈ 4 km | No [1] |
| ERA5-Land (Muñoz Sabater, 2019) | Open | 1981-2021 | Monthly | Global | ≈ 9 km | No [1] |
| CRU TS version 4 (Harris et al., 2020b) | Open | 1901-2018 | Monthly | Global | ≈ 50 km | No [1] |
| HOWNZ | Open | 1910-2019 | Monthly | New Zealand | 1 km | Yes |

[1] An uncertainty grid is provided to aid a user to make a personal uncertainty assessment but it is not in the variables' units.

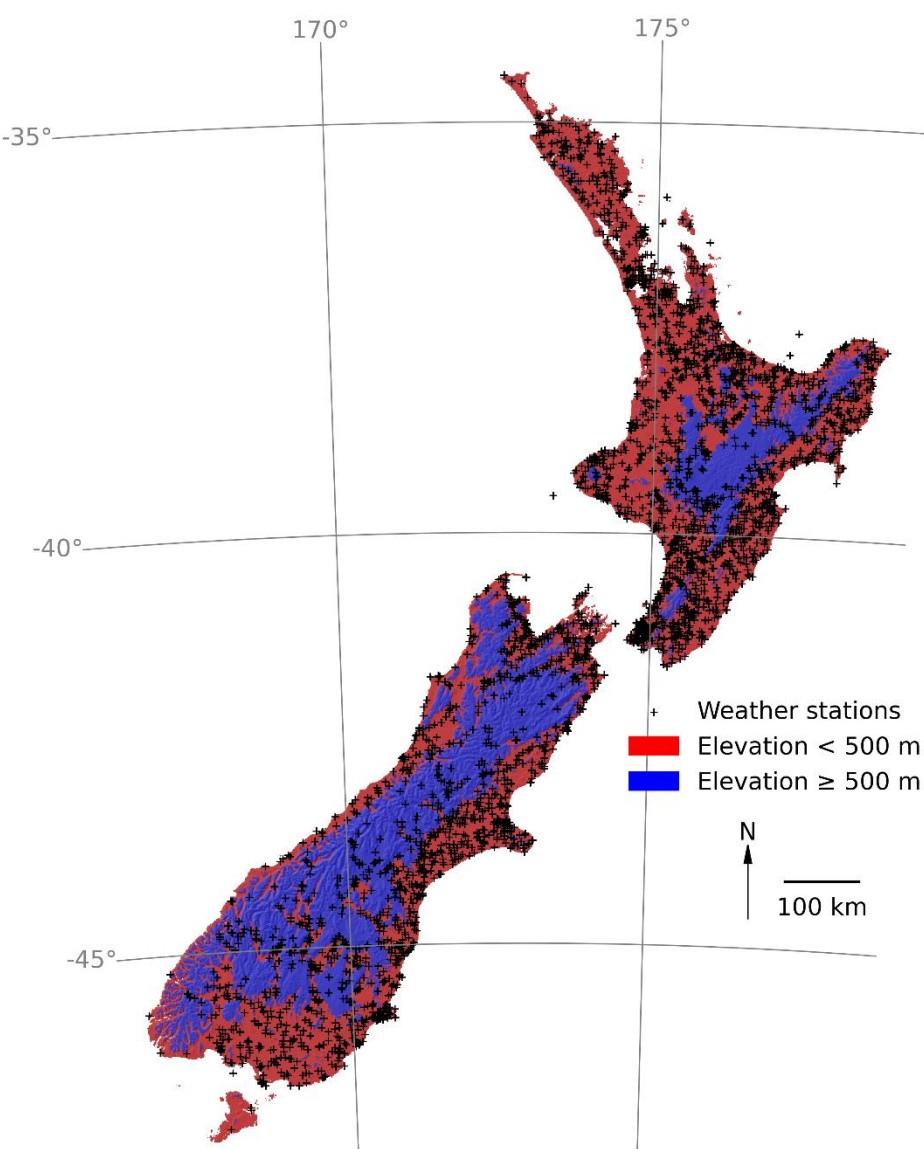


**Figure 1: The topography of New Zealand and the location of all weather stations contributing data.**

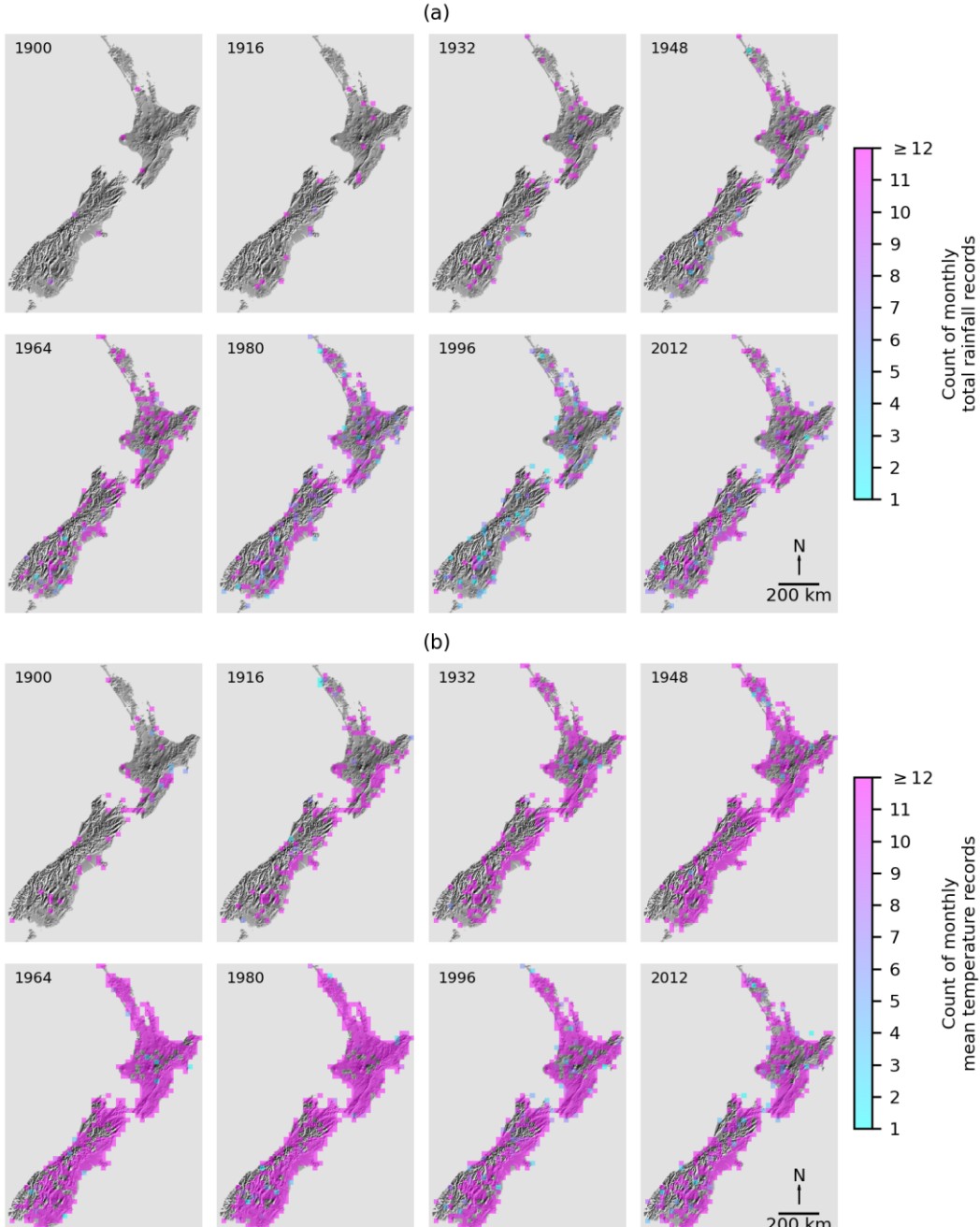

**Figure 2: A time-series of the total number of monthly weather records for each year within 25 × 25 km grid cells for (a) total rainfall and (b) mean air temperature (that are similar to mean daily maximum air temperature and mean daily minimum air temperature). Cells with ≥ 12 records will usually mean at least one weather station is present with records for all 12 months of the year, but in some instances many weather stations may be present resulting in tens or hundreds of records per grid cell.**

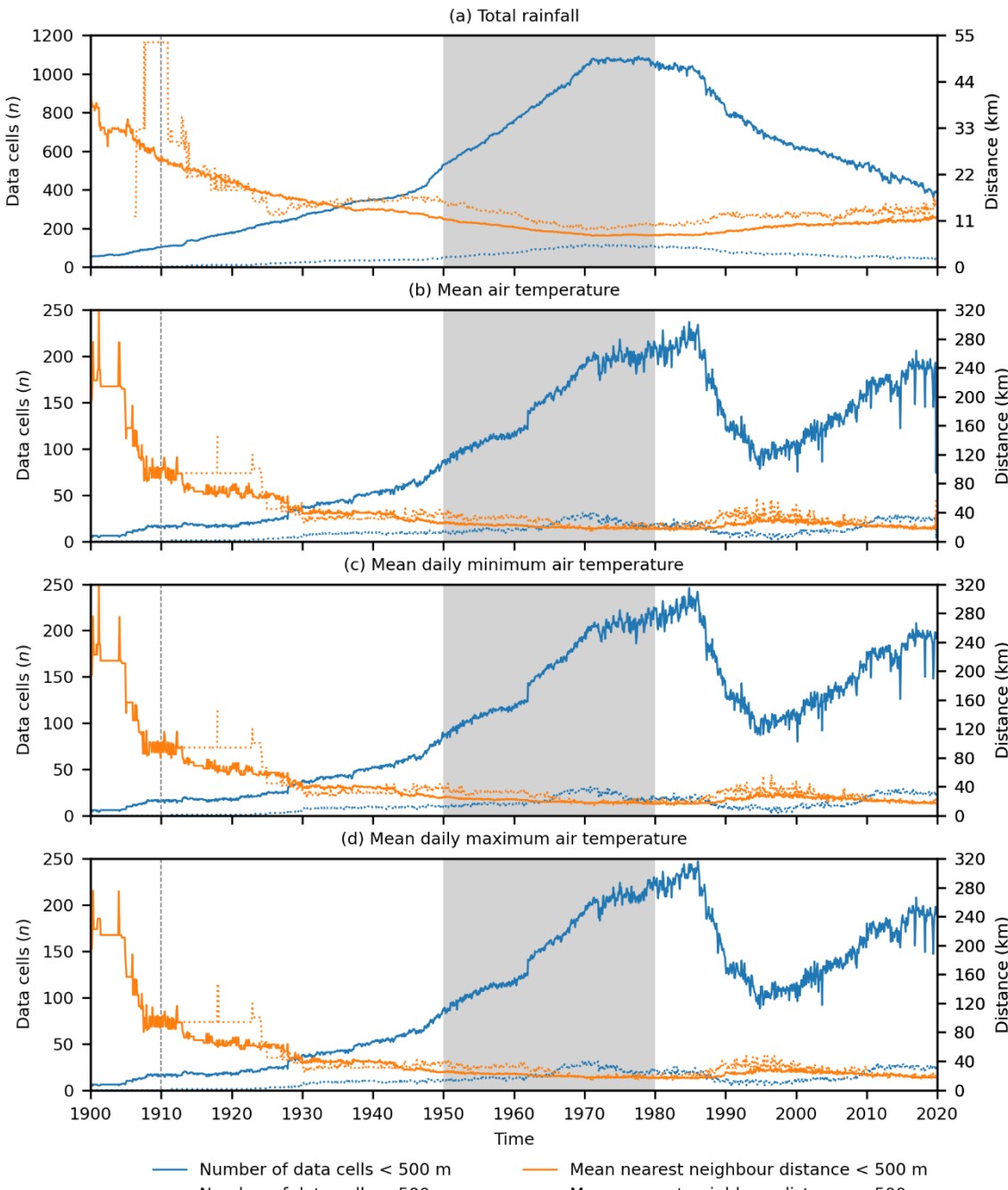

**Figure 3: Monthly time-series of the number of data cells (that relates closely to the number of weather stations) and the mean nearest neighbour distance of data cells contributing to climatologically aided natural neighbour interpolation across New Zealand at < 500 m elevation and ≥ 500 m elevation for (a) total rainfall, (b) mean air temperature, (c) mean daily minimum air temperature, and (d) mean daily maximum air temperature. Missing data indicates an absence of data cells in an elevational category. The grey areas show the period (1950-1980) over which the climatologies used to aid interpolation apply, and the dashed lines indicate the temporal limit of reliable data.**

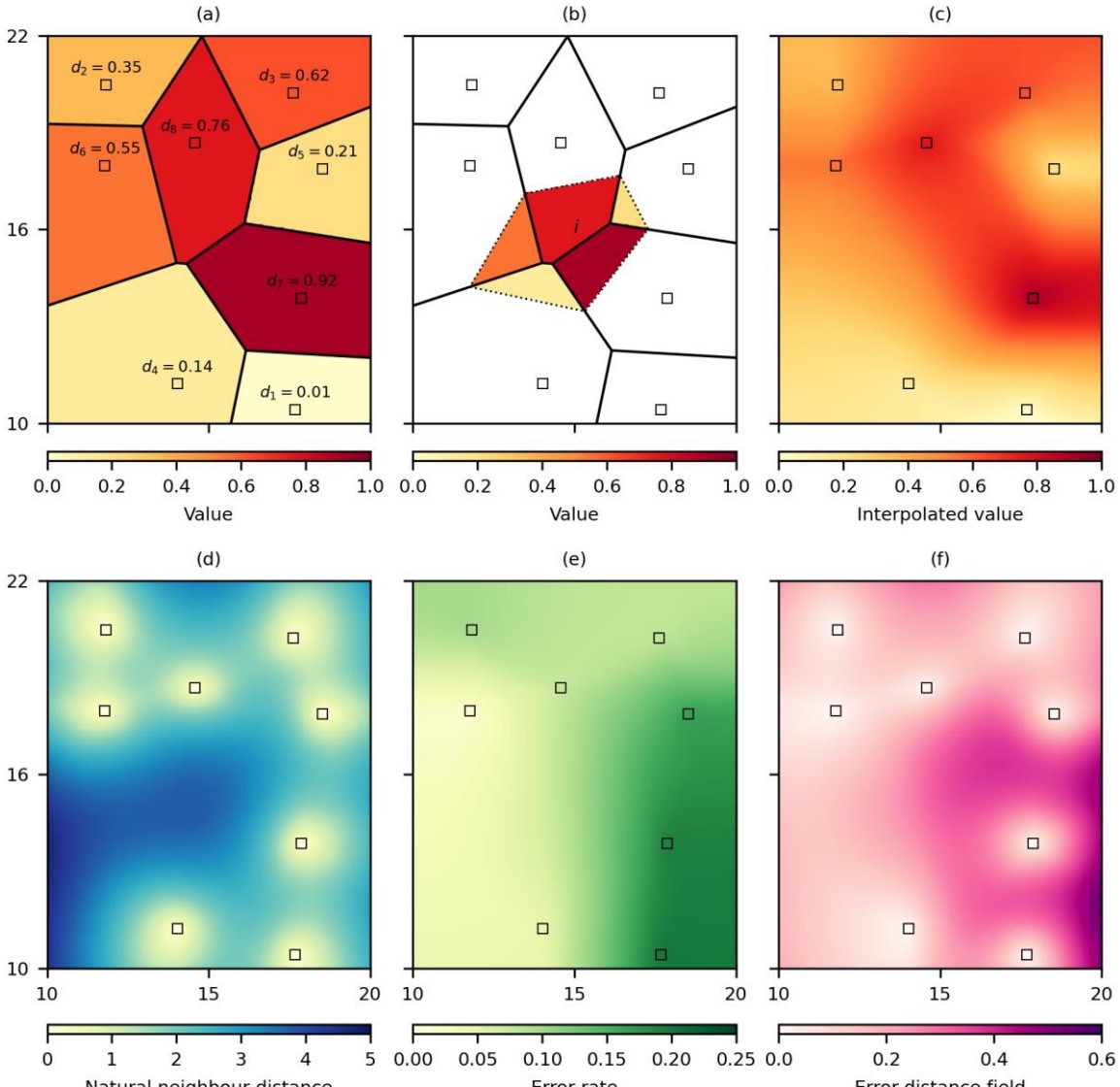

**Figure 4: A hypothetical example to illustrate discrete natural neighbour interpolation with a cross-validation error-distance field.** Beginning with a set of data cells $d_i$, shown as squares, a grid of cells is defined and (a) each cell is given the value of the nearest data cell $d$, (b) for any interpolation cell $i$ the interpolated value is the mean of cell values that are as close or closer to the interpolation cell than a data cell, that (c) when repeated for all grid cells produces a natural neighbour interpolation. Natural neighbour interpolation is also used to interpolate (d) the distances to the data cells, and (e) the cross-validated error rate of each data cell. The interpolation uncertainty is then (f) the cross-validation error distance field that is the product of the natural neighbour distances and error rates (adapted from: Etherington, 2020, https://creativecommons.org/licenses/by/4.0/).

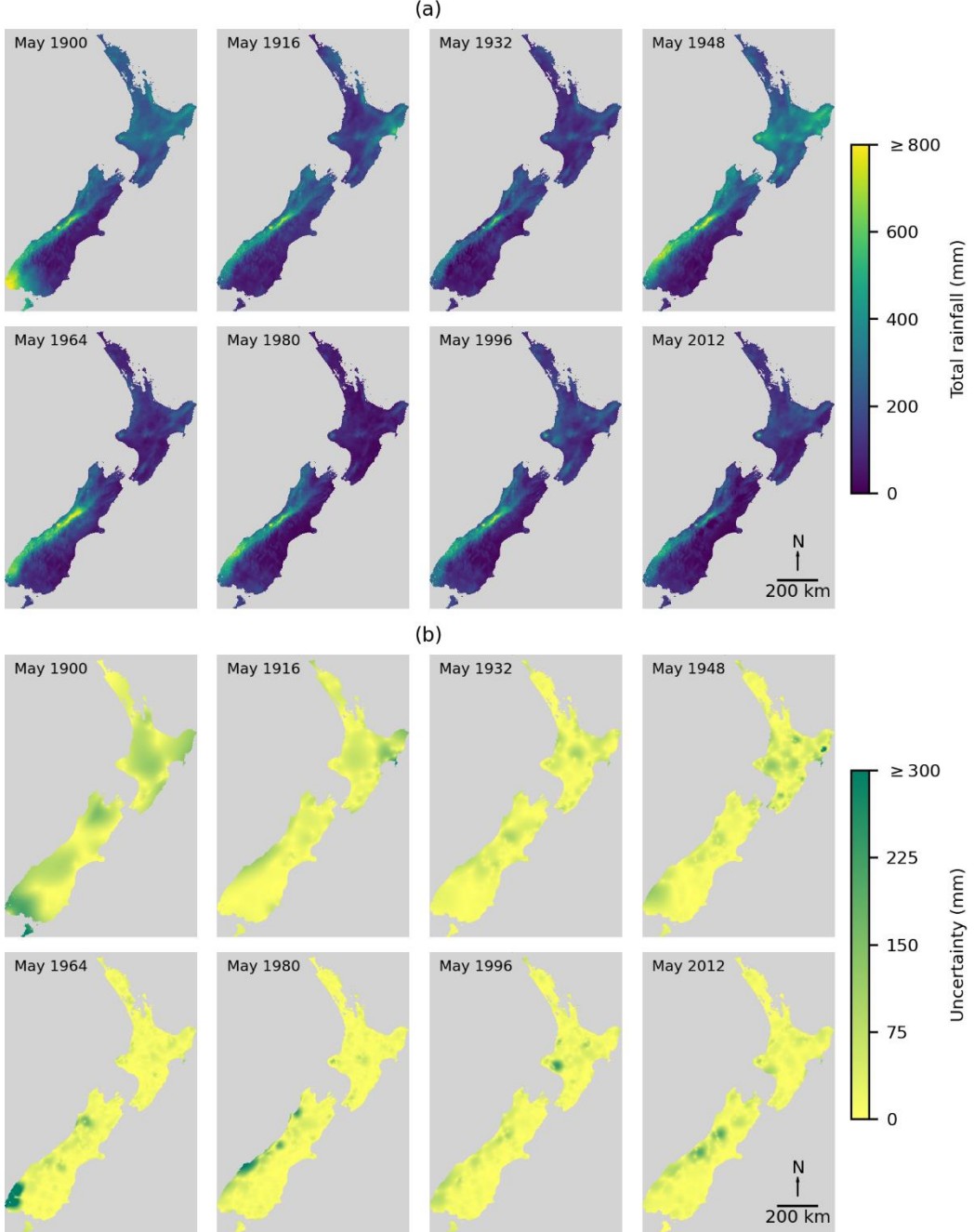

**Figure 5: Example time-series of climatologically aided discrete natural neighbour interpolation of May (a) total rainfall with (b) the associated uncertainty of the interpolated rainfall estimates.**

470

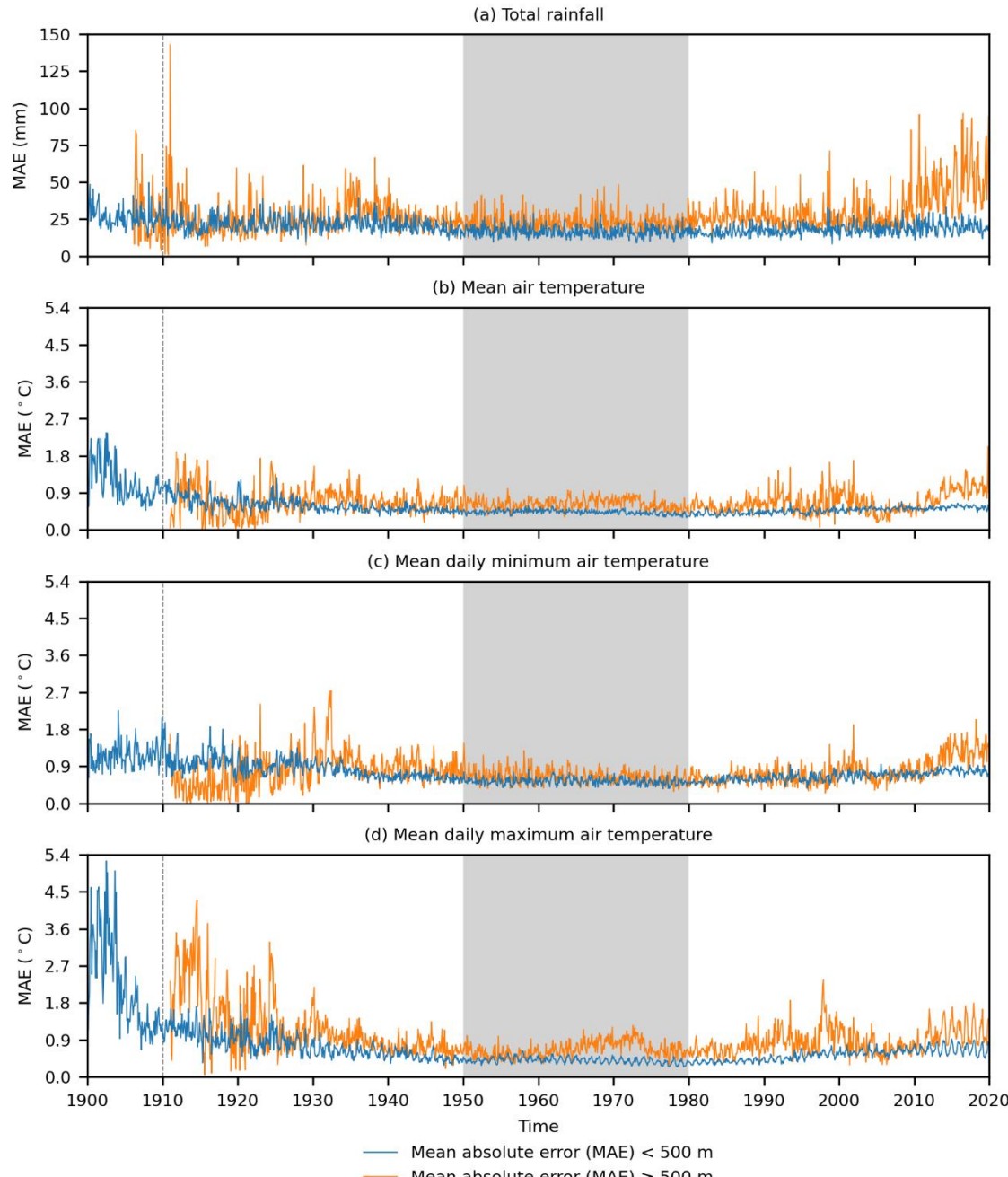

**Figure 6: Monthly time-series of the cross-validated mean absolute error (MAE) of climatologically aided natural neighbour interpolation across New Zealand at < 500 m elevation and ≥ 500 m elevation for (a) total rainfall (mm), (b) mean air temperature (°C), (c) mean daily minimum air temperature (°C), and (d) mean daily maximum air temperature (°C). The grey areas show the period (1950-1980) over which the climatologies used to aid interpolation apply, and the dashed lines indicate the temporal limit of reliable data.**

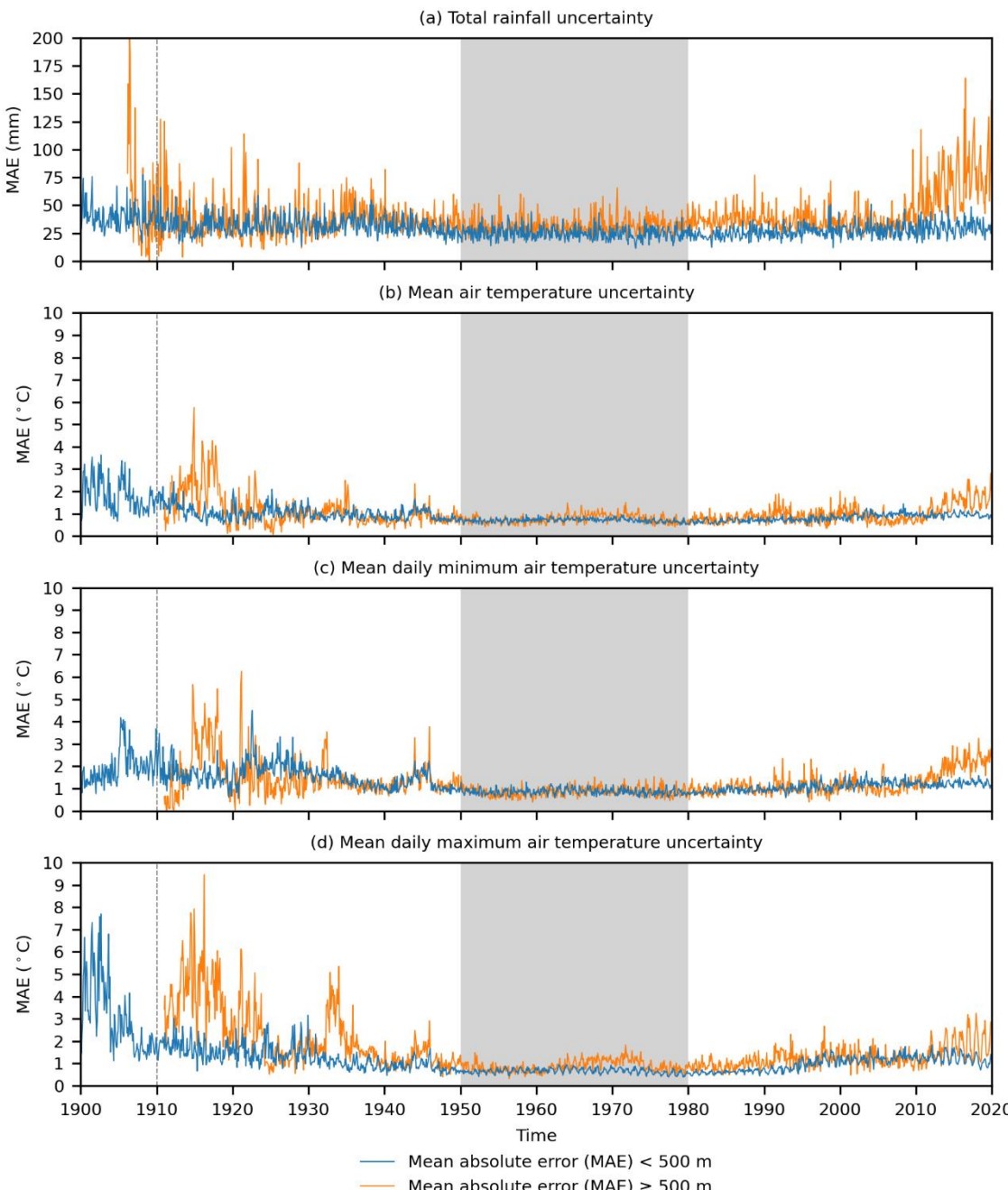

**Figure 7: Monthly time-series of the cross-validated mean absolute error (MAE) of climatologically aided natural neighbour uncertainty across New Zealand at < 500 m elevation and ≥ 500 m elevation for (a) total rainfall (mm), (b) mean air temperature (°C), (c) mean daily minimum air temperature (°C), and (d) mean daily maximum air temperature (°C). The grey areas show the period (1950-1980) over which the climatologies used to aid interpolation apply, and the dashed lines indicate the temporal limit of reliable data.**

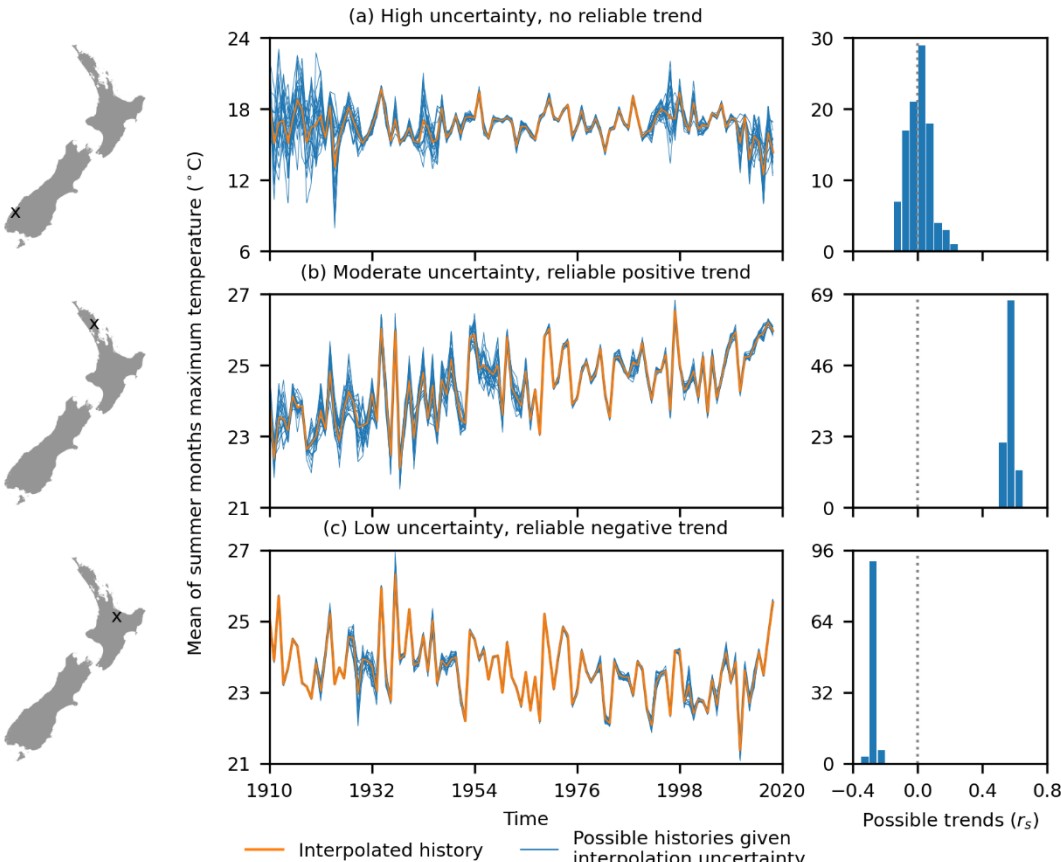

Figure 8: Examples of mean of summer months (December, January, and February) maximum temperature trend analysis that incorporates interpolation uncertainty for three locations in New Zealand. The distribution of trends for the 100 possible histories are shown as Spearman's Rank ($r_s$) correlations between temperature and time for locations with (a) high uncertainty resulting in a wide variety of possible histories with positive and negative trends and no hence reliable trend, (b) moderate uncertainty resulting in some variation in possible histories but all showing a consistent positive trend, and (c) low uncertainty resulting in consistent possible histories and a confident negative trend. For each location, the weather history for the interpolated values is shown along with the first 20 of 100 possible weather histories that are within the range of uncertainty for the interpolated value for each month.

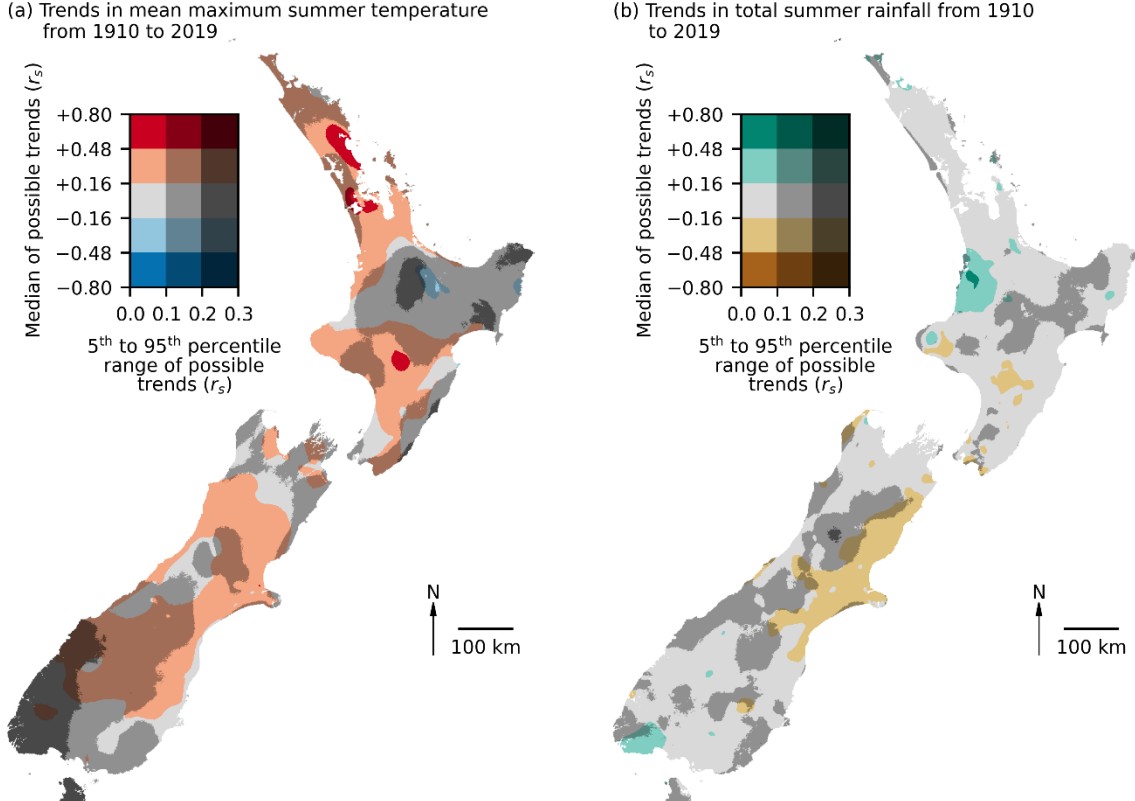

**Figure 9: Spatial uncertainty analysis for the summer months (December, January, and February) trends in (a) mean maximum temperature, and (b) total rainfall across New Zealand. Trends were calculated as a Spearman's Rank ($r_s$) between temperature or rainfall and time from 1910 to 2019 for 100 possible histories that were within the range of uncertainty for the interpolated values for each month. The median $r_s$ value is visualised as a diverging colour scheme indicating positive or negative trends in temperature and rainfall. Uncertainty is visualised by the 5th to 95th percentile range of $r_s$ values as a measure of uncertainty such that darker shades of each colour indicate areas with greater uncertainty.**

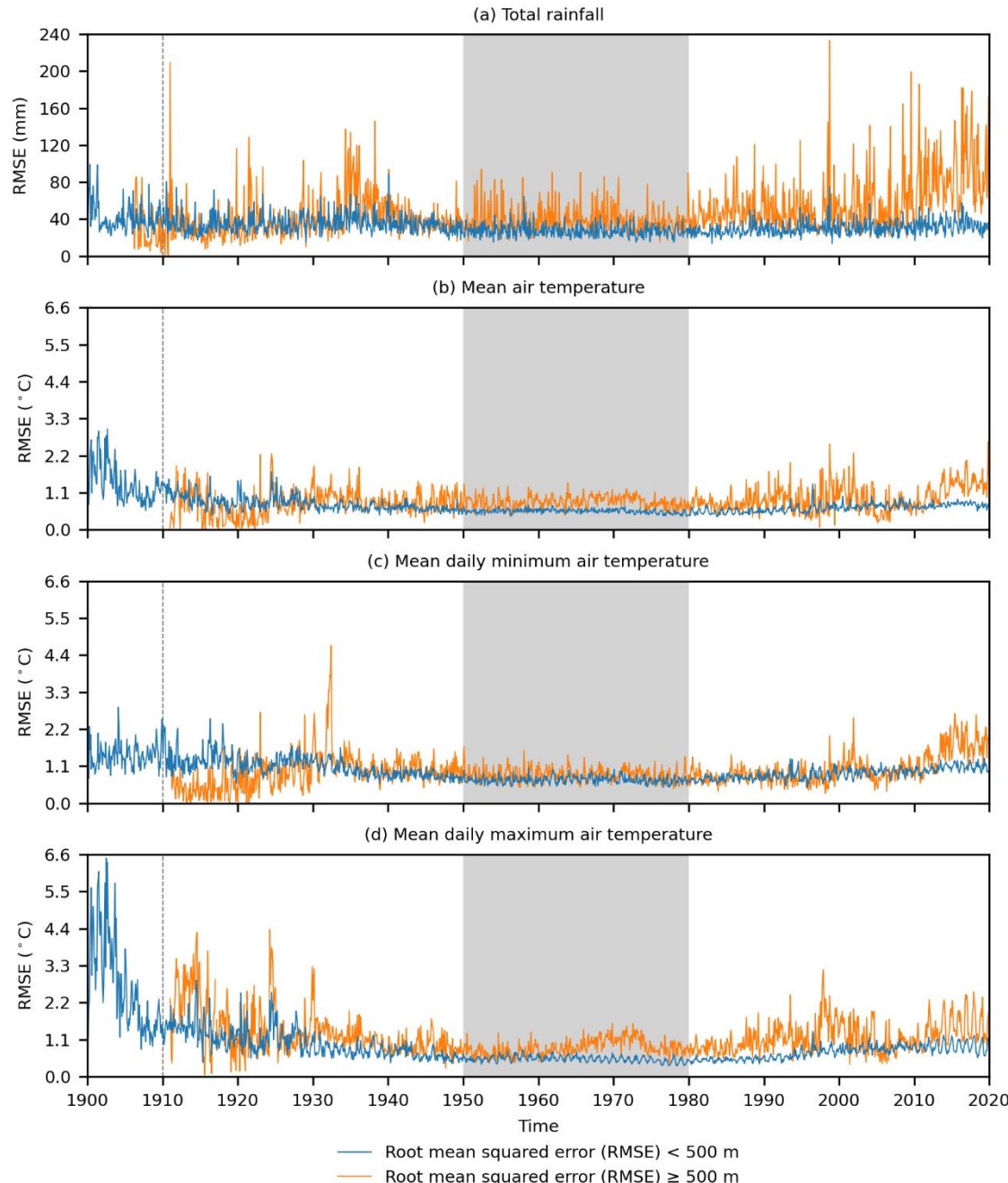

500

**Figure S1: Monthly time-series of the cross-validated root mean squared error (RMSE) of climatologically aided natural neighbour interpolation across New Zealand at < 500 m elevation and ≥ 500 m elevation for (a) total rainfall (mm), (b) mean air temperature (°C), (c) mean daily minimum air temperature (°C), and (d) mean daily maximum air temperature (°C). The grey areas show the period (1950-1980) over which the climatologies used to aid interpolation apply, and the dashed lines indicate the temporal limit of**
505 **reliable data.**

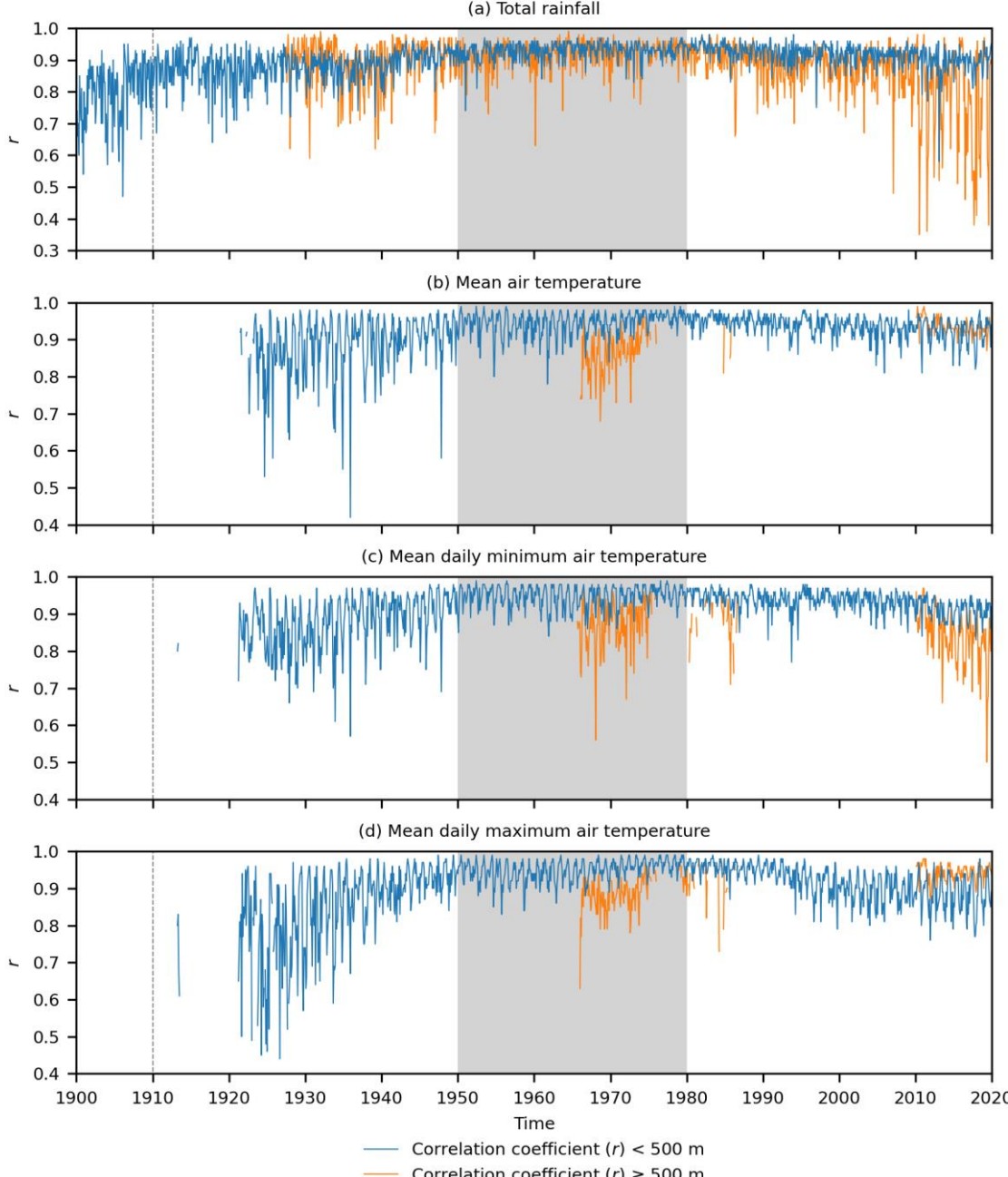

**Figure S2: Monthly time-series of the cross-validated correlation between actual and climatologically aided natural neighbour interpolation values across New Zealand at < 500 m elevation and ≥ 500 m elevation for (a) total rainfall (mm), (b) mean air temperature (°C), (c) mean daily minimum air temperature (°C), and (d) mean daily maximum air temperature (°C). Correlation coefficients are only shown for months where $n \geq 20$ and $p \leq 0.05$. The grey areas show the period (1950-1980) over which the climatologies used to aid interpolation apply, and the dashed lines indicate the temporal limit of reliable data.**

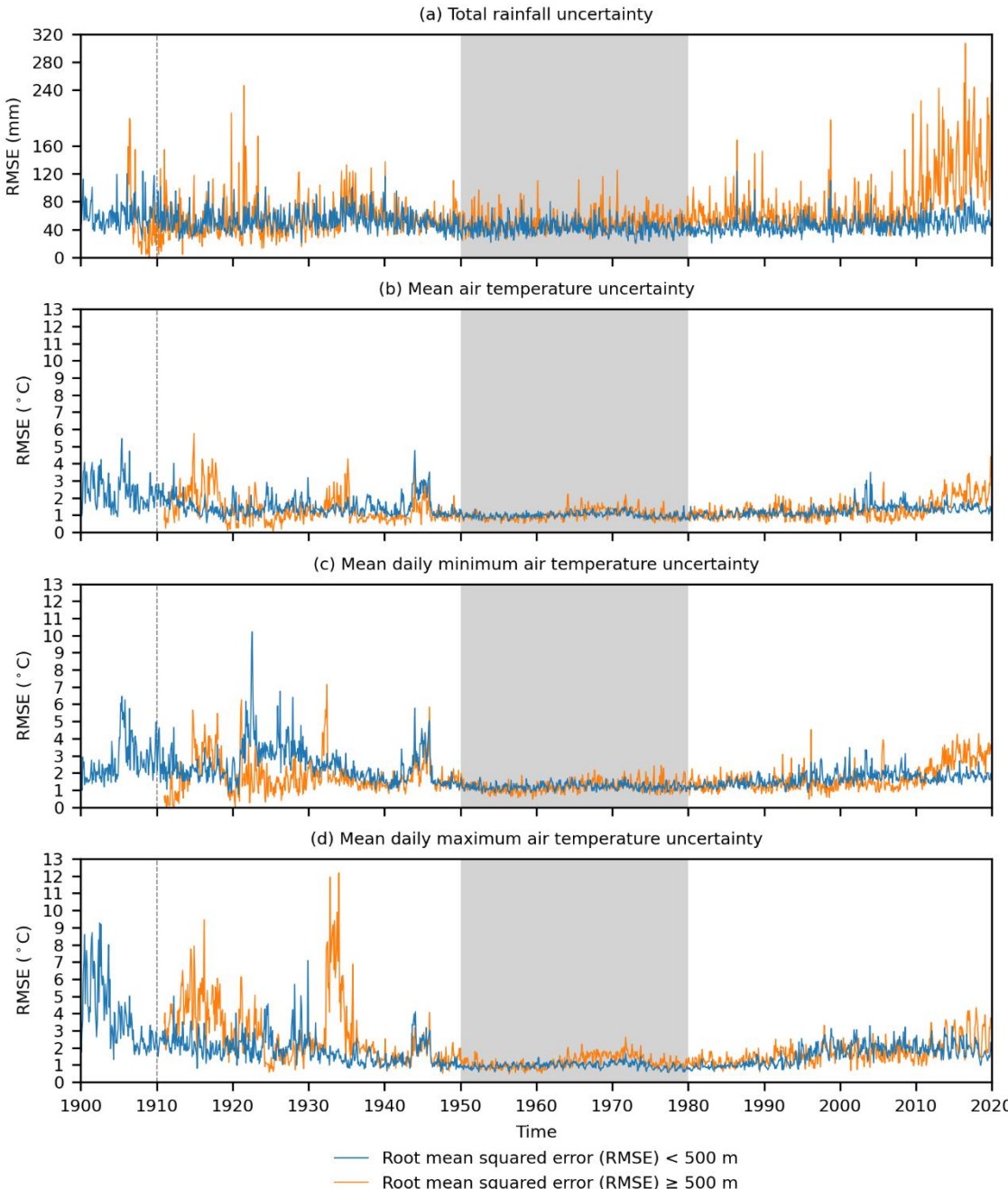

**Figure S3: Monthly time-series of the cross-validated root mean squared error (RMSE) of climatologically aided natural neighbour uncertainty across New Zealand at < 500 m elevation and ≥ 500 m elevation for (a) total rainfall (mm), (b) mean air temperature (°C), (c) mean daily minimum air temperature (°C), and (d) mean daily maximum air temperature (°C). The grey areas show the period (1950-1980) over which the climatologies used to aid interpolation apply, and the dashed lines indicate the temporal limit of reliable data.**

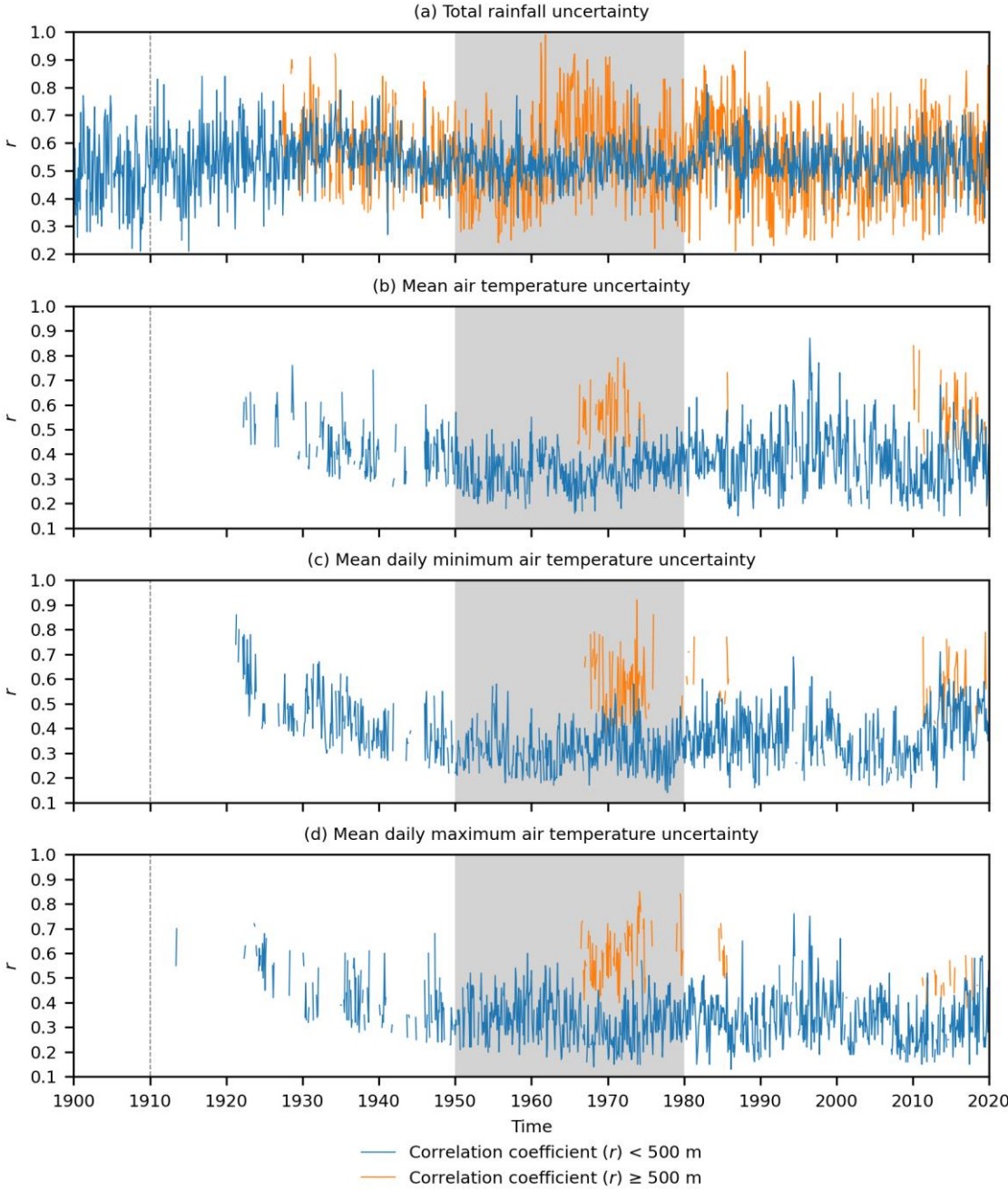

**Figure S4: Monthly time-series of the cross-validated correlation between climatologically aided natural neighbour interpolation errors and estimated uncertainty across New Zealand at < 500 m elevation and ≥ 500 m elevation for (a) total rainfall (mm), (b) mean air temperature (°C), (c) mean daily minimum air temperature (°C), and (d) mean daily maximum air temperature (°C). Correlation coefficients are only shown for months where $n \geq 20$ and $p \leq 0.05$. The grey areas show the period (1950-1980) over which the climatologies used to aid interpolation apply, and the dashed lines indicate the temporal limit of reliable data.**