# Peer review of "HOTRUNZ: an open access 1-km resolution monthly 1910-2019 timeseries of interpolated temperature and rainfall grids with associated uncertainty for New Zealand"

_Earth System Science Data, 2021_

## Author Response (AR1)

Dear Dr Gruber,

Thank you for your patience while we worked through the various reviewer comments. We have now prepared a revised manuscript in which the revisions have been made using track changes so those revisions are clear. We think the comments from the reviewers have resulted in a much stronger description of our dataset, so we are very grateful for the reviewers taking the time to provide their feedback.

We have also provided responses to the reviewers as an enumerated point-by-point list below, cross-referencing between comments where it seems relevant and necessary. Please note the reviewer comments page numbers refer to the original manuscript, while my responses refer to page numbers in the new track changed manuscript.

Hopefully those changes should be clear to you, but if you need me to clarify anything please do let me know.

Once again, many thanks for your time.

Dr Tom Etherington

**CC1**

This is a welcome resource and I agree with the authors that this is likely to have a range of applications. As noted, the applications are limited because the dataset is monthly so extremes in temperature in particular will be masked in the monthly averaging. However, the strengths in the dataset are the long timeseries and the spatial resolution with well-defined uncertainty.

Comment 1: Overall, the paper is well-written but I was interested in seeing a stronger justification for the dataset. For instance, how does the rainfall product improve on work by Andrew Tait (and colleagues), e.g. Tait et al 2012? Is it that this new dataset is freely available, a longer time series, better indication of uncertainty, or perhaps all of these and more? I think a bit more scoping of what is currently available and how this dataset improves on that would be useful. I think open access might be one of the biggest advances here but as a potential end user, I wanted more information around this so I could be certain it was the best dataset (rather than just being what's freely available).

Response 1: As the journal is a specifically open access data journal, we had initially thought non-open access datasets were out of scope. But based on your comment it probably does make sense to include the virtual climate station network (VCSN) data as an option in Table 1 so we have added that information there so that readers can choose the data set that is best for their purposes as while our dataset has a high spatial resolution and longer timeframe, the VSCN provides daily estimates compared to our monthly estimates.

Comment 2: My second suggestion has to do with the choice of climate variables. I agree that rainfall and temperature both highly useful. I suggest adding vapour pressure deficit (or at least relative humidity so VPD can be estimated from temperature and RH). Plants are highly responsive to VPD and this influences plant productivity, water use and survival under extreme events such as drought (see Grossiord et al. (2020) for a review of plant responses to VPD). Because different species respond to drought and VPD in different ways (Volaire 2018), VPD is key to understanding plant species distributions and other factors such as seasonal dynamics in plant moisture (relevant to spatial and temporal patterns of fire risk) and vegetation transpiration and productivity. Beyond

ecology, hydrology is mentioned as a possible application for the dataset. Hydrological models in New Zealand have poor quality estimates of transpiration and this means estimates of streamflow, soil moisture and other hydrological variables are unreliable (because transpiration is a dominant flux in water budgets). Improving the accuracy and reliability of hydrological data will improve our ability to manage water resources, including mitigating against floods and ensuring water is used wisely when scarce. See Whitley et al. (2013), Sulman et al. (2016), Miralles et al. (2019) and more locally, Macinnis-Ng et al. (2013) for examples of VPD as a key driver of transpiration. Including VPD (or RH) as an addition variable in this dataset would increase the range of potential applications of the dataset. While the existing dataset is still valuable contribution, as a potential end-user, I would find the dataset much more useful with this additional variable.

Response 2: This is a good point, and we did not identify other variables that could potentially be produced using our approach. We now specify that the data do exist for solar radiation, humidity, pressure deficit, and wind speed (Lines 271-275). Unfortunately, we are simply not able to do this in this iteration of the data set as NIWA's Cliflo portal for weather station data limits the number of station records that can be downloaded, and we have used up our allowance by completing the four variables we have included. Therefore, such work will have to be left for a later effort that produced a second version of the dataset.

References

Grossiord, C., Buckley, T.N., Cernusak, L.A., Novick, K.A., Poulter, B., Siegwolf, R.T., Sperry, J.S. and McDowell, N.G., 2020. Plant responses to rising vapor pressure deficit. New Phytologist, 226(6), pp.1550-1566.

Macinnis-Ng, C., Schwendenmann, L. and Clearwater, M.J., 2013, June. Radial variation of sap flow of kauri (Agathis australis) during wet and dry summers. In IX International Workshop on Sap Flow 991 (pp. 205-213).

Miralles, D.G., Gentine, P., Seneviratne, S.I. and Teuling, A.J., 2019. Land–atmospheric feedbacks during droughts and heatwaves: state of the science and current challenges. Annals of the New York Academy of Sciences, 1436(1), p.19.

Sulman, B.N., Roman, D.T., Yi, K., Wang, L., Phillips, R.P. and Novick, K.A., 2016. High atmospheric demand for water can limit forest carbon uptake and transpiration as severely as dry soil. Geophysical Research Letters, 43(18), pp.9686-9695.

Tait, A., Sturman, J. and Clark, M., 2012. An assessment of the accuracy of interpolated daily rainfall for New Zealand. Journal of Hydrology (New Zealand), pp.25-44.

Volaire, F., 2018. A unified framework of plant adaptive strategies to drought: crossing scales and disciplines. Global Change Biology, 24(7), pp.2929-2938.

Whitley, R., Taylor, D., Macinnis-Ng, C., Zeppel, M., Yunusa, I., O'Grady, A., Froend, R., Medlyn, B. and Eamus, D., 2013. Developing an empirical model of canopy water flux describing the common response of transpiration to solar radiation and VPD across five contrasting woodlands and forests. Hydrological processes, 27(8), pp.1133-1146.

**RC1**

The current manuscript provides information about a very welcomed open access gridded temperature and rainfall dataset for New Zealand. A 1-km spatial, monthly resolution dataset for the period 1900-2019 has been compiled using weather station data. The weather station data have been interpolated (and likely extrapolated for higher elevations) using a natural neighbour interpolation approach. The data product has been made publically available and will therefore provide an additional data source for a range of researchers. It is important in this context to have a companion paper that provides users with a clear explanation of the value of this new data product.

To help support this effort I have the following suggestions.

Main comments

Comment 3: The reference to the dataset as a "history of open weather" is questionable. The data have been compiled using weather station observations, but a dataset with monthly temporal resolution does not provide information about weather conditions (e.g. L26 and throughout the manuscript). You have created a climatology of temperature and rainfall, but you have not created a "history of open weather" information for New Zealand. I would strongly suggest you carefully define weather versus climate in the manuscript, and be much more careful about the usage of the term weather in the manuscript. I would actually go so far as to suggest you revise the acronym (HOWNZ) – it is appealing but it is not really accurate. There is significant value in creating a new climatology of temperature and precipitation data for New Zealand, but suggesting the dataset will help "facilitate understanding of how long-term weather patterns have been changing and potentially affecting environmental and ecological patterns in New Zealand" (e.g. L53-54) is very misleading. It is not possible to resolve the physical processes associated with weather systems using a monthly climatology, as you indicate yourself in your limitations on L173-75. Please consider changing the focus from the importance of weather to variability and changes in climate.

Response 3: We really appreciate the thoughts provided by the reviewers on our use of terminology, as we simply want to provide the data in as transparent a way as possible. With reflection, we are happy to accept that our grids do not provide information on weather, as there are definitions of weather that relate to conditions from a few hours to a day. That said, we're also not happy to refer to our time-series of monthly rainfall and temperature grids as climate estimates as the reviewer suggests. We define climate and climatology at the beginning of the Introduction section to relate to a 30-year period (see Lines 27-28), so to also use climate to mean a monthly period is we think potentially confusing (especially as there was already some confusion around the differences between a climatology's 30-year average for each month and a time-series' monthly averages over 30 years: see Comments 23 and 38). Essentially, we do not think there is clear terminology for exactly what meteorological or climatological entity monthly time-series data provides – and this might explain why other projects such as WorldClim 2 use the phrase monthly weather (see: https://www.worldclim.org/data/monthlywth.html). Having reflected on this, we think the most pragmatic solution is not to try and define what our monthly rainfall and temperature grids should be interpreted as. Instead, we will simply refer to them as "monthly rainfall and temperature grids" and avoid labelling them as either "weather" or "climate". Therefore, we have made changes throughout the revised manuscript and in the supporting data and code archives to reflect this. Of course, this means we need to drop the W for weather from the HOWNZ name. While working through other revisions it has become more apparent to us that we may have under-emphasised just how novel the uncertainty data is. Therefore, we have adopted a new name of: HOTRUNZ, a History of Open Temperature and Rainfall with Uncertainty for New Zealand. This change has been made throughout the revised manuscript and in the supporting data and code archives.

Comment 4: Section 2 provides information about the data sources and there are a number of figures in the manuscript that detail the number of weather records available for the interpolation (e.g. Fig. 1 and Fig. 4). However, additional detail in a (supplementary) table that provides more explicit information about the data sources – the number of stations and how they compare to previous reconstructions - would be welcomed by data users I believe. For example, a ranking or list that provides the primary or most important data sources would be very useful – it is presently quite black box – with just a reference that data were sourced from New Zealand's National Climate Database (L60). There is very little information about the quality controls of the input data (if any) or any challenges that were faced, with the reference to some of these complexities lacking important detail (e.g. L64; "there were some [in]consistencies", L64; "much more rainfall than temperature data", L65; "there was always less data in mountainous interiors". The compilation of archived weather station data is a massive task – even if primarily sourced from a known database, which you have taken on admirably. I would strongly recommend trying to provide additional detail about the quality and quantity of the data used for each year and region (or elevation band). It will be warmly welcomed by readers and future users of the data.

Response 4:  I wonder if there is some confusion here as we only used data from the one database. Therefore, rather than say we "compiled all reliable data", which we acknowledge could be read as suggesting that we have pulled together data from various sources, we now say we "queried the database for data" (Lines 66-67) to try and make it clear that all the data comes from a single source, and that the credit for the compilation of archived weather station data belongs to the creators and maintainers of that database.  Our quality controls were based entirely on the information provided by that database, and we have expanded the description of how we filtered data in the hope that readers will better understand the quality of the data being used, for which we feel we have set a very high bar (Lines 69-75).  We also now provide information on the quantity and spatial pattern of data by elevational band (see Figure 3).  We used 500 m elevation as a breakpoint to be consistent with Tait et al. (2012) as these authors noted that nearly all the weather stations in New Zealand occur below 500 m, a limitation we emphasise when introducing the weather stations' data (see Lines 77-82 and Figure 1).

Comment 5: Section 5 provides information for users about how to use the uncertainty data that has been generated for analytical workflows in the future. This will be appreciated by readers and is quite useful, but there is still a need to show in some way how the uncertainty generated from the validation compares to independent observations or to another gridded data set (e.g. VCSN data). There are well known deficiencies in the VCSN data (e.g. Tait et al., 2012; Jobst et al., 2017), especially at higher elevations and some demonstration about how the new data set compares should be considered. At the very least a warning or some comment about the need to test the new data should be included, and acknowledgement of the efforts by others to address some of these uncertainties should be considered. There is very little reference to other research or efforts to address these problems in the current manuscript. A quick comparison between VCSN and the new data set would be quite revealing and provide some indication of how they compare, and what users might consider looking out for when using the newly compiled data.

Response 5: Providing an indication of how well the uncertainty works is an excellent suggestion, and with hindsight is something we really should have included initially, so we appreciate the suggestion.  However, we do not think comparing the uncertainty estimates to another gridded data set is the best approach, as any differences could either be a result of our uncertainties being wrong or the comparative grids being wrong.  There are also some particular issues with using the VCSN data as they are not open access as required for data used in an Earth System Science Data paper. Furthermore, differences in the spatial (1 km versus 5 km) and temporal (monthly versus daily)

resolutions means that both datasets would have to be aggregated before comparison, which means that we are not really comparing the datasets in their true form anymore and we would not be confident that any conclusions from such an analysis could be reliably transferred to the actual data. Therefore, we think the best approach is to extend the cross-validation analysis to evaluate how well the uncertainty estimates capture the actual errors. Details of this approach and the results can be found on Lines 199-205, Figure 7, and supplementary Figures S3 and S4. Regarding known challenges of interpolating at higher elevations, we have included some references to previous work demonstrating this (Tait and Macara, 2014; Tait et al., 2012) and have broken our evaluation down by elevational range, please see Comment and Response 4 for further details.

Specific comments – including editorial suggestions

Comment 6: L49: Add one-character spacing between authors listed. This occurs elsewhere in the manuscript.

Response 6: We are using the EndNote style file provided by the journal, so I'm afraid we can't change this.

Comment 7: L101: Remove "that areas" after in areas.

Response 7: Done, thanks for spotting.

Comment 8: L110: Suggest changing the wording to "uncertainty can be high in regions where weather data are available".

Response 8: Done, again, thanks.

Comment 9: L126-27: Be specific when referring to "weather data availability becomes extremely limited" – see main comment 2. I believe further detail about the quantity and quality of the input data would be welcomed by readers.

Response 9: We have added some text here to explain in more detail what we mean by "extremely limited" which refers to the number of data cells in each month (Lines 191-192). Please refer to Comment and Response 4 for how we have included more information around the quality and quantity of the data.

References

Jobst, A. M., Kingston, D. G., Cullen, N. J., & Sirguey, P. (2017). Combining thin-plate spline interpolation with a lapse rate model to produce daily air temperature estimates in a data-sparse alpine catchment. International Journal of Climatology, 37(1), 214-229. doi: 10.1002/joc.4699

Tait A., Sturman J., & Clark M., (2012). An assessment of the accuracy of interpolated daily rainfall for New Zealand. Journal of Hydrology (NZ), 51(1), 25-44.

**RC2**

The authors describe a monthly climate data set for New Zealand that is based on an interpolation of station data. The use a relatively simple interpolation approach – natural neighbour interpolation, and present monthly fields. Overall the paper is a welcome contribution and the data set will be useful for different applications. However, there are some comments that should be considered in the revision.

Comment 10: Why is this data set referred to a "Weather Data"? In my view this is a climate data set.

Response 10: We no longer refer to our data as weather data, please refer to Comment and Response 3 for a fuller description of the changes made.

Comment 11: Not much is said about the underlying station data. Are these homogenised records? Was a breakpoint detection performed? (And how would inhomogeneities affect the result?)

Response 11: We have provided additional details about how we queried and filtered weather station data (Lines 69-75). We did not perform any kind of breakpoint detection or homogenisation, and have trusted that the experts who have complied this data have done so correctly, and we think this is reasonable as we set the highest standard we could for the data we extracted from the database, and did not use any records flagged as potentially unreliable.

Comment 12: Perhaps a minor point, but the authors speak interchangeably of precipitation and rainfall. What about snow? This should be clarified.

Response 12: We are a little confused as we only used "precipitation" once in the paper in the caption for Table 1, but you are right to point out this inconsistency, and this has been corrected to rainfall, thanks.

Comment 13: The authors indicate the number of cells with station data – perhaps also some other measures (e.g., average inter-station distance) would help in the comparison with similar products from other regions.

Response 13: This was a superb suggestion, thank you! We have now calculated the mean nearest neighbour distance between data cells for each month (see Figure 3), and we can see that although the number of data cells fluctuates over time, in general the mean nearest neighbour distance has remained reasonably consistent, at least in more recent times. This would indicate that while the number of weather stations has fluctuated, the weather station network has maintained similar levels of spatial coverage (Lines 135-136). This consistency of spatial coverage might well explain why the interpolations don't appear to be sensitive to the number of data cells (Lines 184-185).

Comment 14: The authors interpolate in anomaly space, but this implies that product used is fully consistent with observations. Were the station data compared to the climatology in the 1950-1980 period? (There could be biases, e.g., due to different altitudes in the underlying climatology, or other biases, urban effects etc.).

Response 14: We did not compare weather station data to the climatologies, so we are certainly working on the assumption that the climatology is suitable to the task. We believe this is a fair assumption as the climatologies are built from the same weather data sources, albeit they are a subset thereof. If there are any inconsistencies these would present themselves through increased

errors and hence greater uncertainties, which while obviously not ideal, we feel we will have recorded sufficiently well for users of the data to be able to recognise problematic areas.

Comment 15: The description of the method is not fully clear. For instance, the description of Fig. 2b ("mean of grid cell values that are as close or closer ... than a data cell") implies a circular region, whereas this is a Voronoi polygon. This could be addressed by adding, to Figure 2b, also the observation locations and the polygon outlines.

Response 15: Thanks for the suggestion, we have done this as requested, please see the new Figure 4.

Comment 16: I had difficulties with the description on line 98f. I struggled with the term "error rate", which is defined as " the ratio of the cross-validated absolute error to natural neighbour distance". This is then multiplied again with the natural neighbour distance (by which it was divided in the previous step). I struggle because the error rate is interpolated from the station cell, right? Isn't the natural neighbour distance zero at the location of a data cell (so, the error rate is undefined), or if it is 1 km (grid cell length), then the error rate is numerically the same as the cross-validated absolute error. I understand that the error rate must be in K per m and the "error-distance field" in K.

Response 16: Apologies that we didn't explain this sufficiently clearly.  A full and detailed description of the uncertainty method would require a lot of space and would simply be repeating information in Etherington (2020), and we don't think that is appropriate here.  But we also don't want to just simply "refer you to the paper" for more details as this is unhelpful to the reader.  So rather than try and replicate the technical description in Etherington (2020), we have inserted what we hope is a more intuitive description of the essence of the approach (Lines 149-155).  We hope that this provides sufficient generic context to make the more technical text that you queried more understandable.

Comment 17: Fig. 2 shows the interesting feature that the error can become small despite a large distance in regions between two stations with good cross-validation skill and no other station nearby.

Response 17: Yes, that is correct and is an assumption of the uncertainty calculations.

Comment 18: It is certainly an asset to provide the error as indicated in the paper. For the evaluation, it would however be interesting to also know some other measures (that do not necessarily need to be interpolated). Hofstra et al. list several.

Response 18: Thanks for the suggestion, we have now also included RMSE and correlation metrics to compliment the MAE data (Lines 195-197, and supplementary Figures S1 and S2).

Comment 19: The methods does not make use of topographical information. Other studies (which the authors cite, e.g., Hofstra et al.) show that different methods perform differently well in mountainous terrain. New Zealand must be a prime example, yet this topic is only touched upon very briefly.

Response 19: I am slightly confused as to whether the lack of topographical information use relates to the interpolation method itself or more specifically our error analysis.  Regarding use of topography in the interpolations please refer to Comment and Response 28.  Regarding performance in mountainous terrain, we have now broken down our interpolation evaluation in land above and below 500 m such that the effects of mountains can be assessed.  We used 500 m as a

threshold to be consistent with work by Tait et al. (2012) who also assessed the ability to interpolate rainfall in New Zealand.

Comment 20: In the examples used on Section 5, it might be interesting to say a few words about autocorrelation of errors (and spatial covariance).

Response 20: Sure, we've added a couple of sentences to highlight the spatial pattern of the uncertainty (Lines 243-247).

**RC3**

The data set is novel. However, the methods, materials, and output uncertainties are not described in sufficient and clear detail, and the interpolation of monthly mean meteorological data has occurred without any consideration of key geographical aspects (elevation and proximity to the coast). This severely compromises the utility of the data. The data set is accessible via the given identifier. The data (presumably numbers that people can use in calculations) are provided in Tag Image File Format (*.tif files), which have long been used primarily for images. No guidance is given on how to access or extract numerical values from these, which also limits the dataset's utility.

Of particular concern to me are

• the interpolation of meteorological data in mountainous regions without regards to changes in elevation between observations,

• the lack of precision with language in statistical settings (e.g. "reliable", "uncertainty") and/or specific meteorological meaning (e.g. "dynamical"),

• the re-use of published figures without explicit attribution.

MAJOR COMMENTS:

Comment 21: WEATHER & CLIMATOLOGY DATA (lines 61-3): What do you mean by "reliable" and how did you decide the data were reliable? What is "sub-kilometre spatial precision", and how was the precision of the data determined? Did you use all the data you obtained? If not, what was your decision process and the results? For the monthly mean air temperatures, how were those calculated and was the calculation the same across stations and time? (i.e. was it an average of hourly data, an average of max and min data, or something else?)

Response 21: We have expanded our description of the source of and the selection criteria we used for the monthly rainfall and temperature data (Lines 66-75). All these decisions were based on information from the climate database, so we did not quantify the quality of the data ourselves, rather we used quality flags in the database to select the data most reliable for our needs. We did not create the monthly values ourselves, as these were pre-calculated within the database we queried, so we have tried to make that clearer too.

Comment 22: WEATHER & CLIMATOLOGY DATA (lines 63-67, Figure 1): There should be latitude, longitude, and some sort of km scale on all these maps. More information about the "mountainous interiors" is needed, specifically the horizontal and vertical variability. A single, high-resolution, easily read map for the entirety of New Zealand would be a fine addition; the greyscale underlays in Fig. 1 seems to show hills as well as mountains throughout the country.

Response 22: The suggestion of an introductory figure giving some context about New Zealand is a good one. With hindsight we may have assumed too much of our readers given our local knowledge. Therefore, we have created a new Figure 1 that shows the latitude and longitude across New Zealand and shows the topography in more detail. We have used 500 m as a cut-off to define mountainous interiors to be consistent with other work (Tait et al. 2012) evaluating the interpolation of rainfall in New Zealand. Given the new Figure 1 gives the latitude and longitude context, we have not included the latitude and longitude on other figures as we feel that is redundant information that will only serve to clutter the figures; we have added scale bars to all the maps as that is a useful suggestion for readers to allow them to better understand the distances involved.

Comment 23: WEATHER & CLIMATOLOGY DATA (lines 68-69): "Reprojected and aggregated to 1-km$^2$ grid cell resolution" needs more explanation. Also, how did you ensure that these data were never duplicating the NIWA data? Or did you use these as the climatology for your "climatologically aided interpolation"? If so, that's unclear when reading this; you should clarify here that the climatological data will be used in the calculation of anomalies as part of the interpolation process.

Response 23: We can see how this could be confusing. We have moved the text introducing the concept of climatologically aided interpolation ahead of this section to remind readers what a climatology is and how it is relevant to our interpolation approach. We also reworded the description of the climatologies to try to avoid any confusion about what they were, and removed the terms "reprojected and aggregated" as while these are common GIS terms, they are unnecessarily specific for a more general readership (see Lines 100-103)

Comment 24: INTERPOLATION WITH UNCERTAINTY (lines 72-79): What are the advantages and disadvantages to climatologically aided interpolation? Why not just interpolate the data, rather than calculating anomalies and interpolating them? It seems to me that the calculation of anomalies might add its own (possibly spatial) error potential.

Response 24: As we stated in the paper, we used climatologically aided interpolation as it has been shown to be a successful approach both in New Zealand and beyond (Lines 89-92), but I have now included some key advantages of climatologically aided interpolation (Lines 96-100).

Comment 25: INTERPOLATION WITH UNCERTAINTY (lines 83-5): What do you mean that natural neighbour interpolation will "retain the original data values in the interpolated grid"?

Response 25: We have clarified that this means that in locations where there are input data values the interpolated grid will retain the values of those input data (Line 114-115).

Comment 26: INTERPOLATION WITH UNCERTAINTY (lines 91-102): This paragraph serves to highlight my confusion about how you are using the word "grid" and "cell". As a meteorologist who has worked with numerical models of the atmosphere and their output, when I read about data being on a "grid" I think of a mesh of evenly spaced points in (x,y) or (x,y,z) space, where typically x=east-west distance, y=north-south distance, z=vertical distance, and âx=ây. I'm unfamiliar with the notion of cells, but would guess that they are perhaps the contents of a grid box.
How do you "define a grid of cells"? What is a "data cell"? What is a "grid cell"?
Also, what areas are "harder to interpolate accurately", and why?

Response 26: Regarding definition of grid and cells, please refer to Comment and Response 39. Regarding data cells, please refer to Comment and Response 43. Regarding areas that are harder to interpolate to, we have extended the sentence to clarify that these are areas with greater spatial heterogeneity and sparser data and have inserted a supporting reference (Lines 160-161).

Comment 27: INTERPOLATION WITH UNCERTAINTY (Figure 2): Figures 2a, 2b, and 2c appear to be identical to Etherington (2020) Figures 1c, 1e, and 1f, respectively. Figure 2d appears to be identical to Etherington (2020) Figure 2c. Figures 2e and 2f appear to be Figures 3a and 3b from Etherington (2020), with the annotations on the 8 small squares removed from the former and the color scale changed in the latter. However, there is no citation or acknowledgement of the re-use and (in some cases) adaptation. While Etherington (2020) is distributed under a CC-BY 4.0 license, that license's terms require "You must give appropriate credit, provide a link to the license, and indicate if changes were made." (https://creativecommons.org/licenses/by/4.0/).

Response 27: Thanks, and yes, this is an oversight on our part, so a note has been added to the figure caption (Line 484).

Comment 28: INTERPOLATION WITH UNCERTAINTY (lines 91-102): The manuscript touts the horizontal resolution of the new data set, but says almost nothing concrete about space in the creation of the data set. What is the range of distances over which interpolation occurs? How does it vary across New Zealand and across time? What are the impacts? And most importantly: What about elevation and distance from the ocean? Elevation has *major* impacts on precipitation and temperatures; nearness to the coast, even in the absence of any elevation changes, has major impacts on temperatures. Yet the interpolation scheme appears to ignore geography. This is especially astounding for 2 reasons: the NZEnvDS (McCarthy et al. 2021) contains a wide range of geographical information that would affect the weather measurements, including distance to coast and various terrain data, and the repeatedly referenced Willmott and Matsuura () compares a planar temperature interpolation with a digital elevation model (DEM) assisted interpolation.

Do geographical factors ever constrain your interpolation? If so, how? If not, why, and from a meteorological rather than a statistical point of view what are the implications?

Response 28: Regarding distances over which interpolation occurs, readers can get a sense of this via the new Figure 1 that shows the distribution of all weather stations contributing data, and via the new Figure 3 which shows the mean nearest neighbour distance of data over time.

Regarding geographical factors, yes, geographical factors most definitely constrain the climatology aided interpolation via the climatology. We acknowledge that we did not make this clear, therefore we have added a note to that effect when describing the advantages of climatology aided interpolation (Lines 98-99). That said, we acknowledge that the geographical factors included via the climatology are done so indirectly and that we failed to provide any information about that. Thus, we have added a brief description of how the climatologies were created, and the geographic variables used in this process which includes elevation, and in the case of rainfall an east-west topographic protection variable (Lines 103-108). You are correct that a multitude of factors may affect temperature and rainfall patterns and that more complex methods would be able to factor into the interpolation. However, for our application we did not feel we could use more complex methods given the paucity of data at higher elevations and further back in time, and evidence that in such situations simpler models may perform better than more complex models – as was also our experience from some earlier exploratory attempts that used kriging and geographically weighted regression in combination with predictors such as elevation. Therefore, we have clarified our decision logic (Lines 120-124), and otherwise rely on our evaluation metrics, which show the method performs sufficiently well to be useful especially given the paucity of other options (Table 1).

Comment 29: INTERPOLATION WITH UNCERTAINTY (lines 104-113): The analysis of this section is trivial, and boils down to saying that the maps show horizontal differences in both values and uncertainty, and they change in time. "Clear differences" are not necessarily correct; neither are calculated gridpoint values of uncertainty, and their "dramatic changes". Why do you interpret "dynamic uncertainty" (by which you seem to mean "changing uncertainty" rather than "uncertainty connected with meteorological dynamics") as being due to sparse data rather than meteorological incompatability of data collected in highly variable terrain? What do you mean by, and do you have any evidence of, the "spatial variability of individual monthly weather patterns"? (Weather tends to operate on timescales of minutes to a few days; see https://glossary.ametsoc.org/wiki/Weather.)

Response 29: Regarding the point about the timescales of weather, we accept that the term weather is inappropriate for our monthly data – please see Comment and Response 3 for further discussion. We respectfully disagree that showing this information is trivial, as it supports our contention that global error estimates such as MAE are of limited use, as they do not provide any indication of where uncertainty is greatest, and that the location of the greatest uncertainty could be different for every month. We fully acknowledge that in addition to data limitations variability in terrain will make the interpolations harder. We have added additional text to the manuscript to clarify these points (Lines 168-176).

Comment 30: INTERPOLATION WITH UNCERTAINTY (overall): There is no discussion of the uncertainties of the source data, and it is not clear whether the "uncertainty" provided in the gridded data represents purely computational error (i.e. is attributable only to the interpolation) or includes measurement error. This needs to be explicitly discussed in the paper and clearly identified in the data set.

Response 30: This point about quite what our uncertainty seeks to capture is well taken, and we have clarified that our uncertainty relates purely to computational error associated with our interpolation, and does not include uncertainty associated with measurement error (Lines 259-269). We did set a high bar for station data to be included in our analyses, and we also now more explicitly state that criteria for inclusion (Lines 69-75).

Comment 31: INTERPOLATION VALIDATION (lines 121-125): What other interpretations are there for the consistency of the New-Zealand-average MAE for all weather variables over time, and why do you reject them? Given the changes in measurement techniques over time, and the extreme difficulty of measuring preciptation at any time, I find the consistency surprising. What does the distribution of MAE across New Zealand look like throughout time? (This is especially important since one of you major points is that your data are better than others because you have gridpoint values of uncertainty.) What does the distribution look like a few particular times at different locations?

Why do you expect the MAE to be lowest during the period of the climatological data used to calculate anomalies? Why does that fact not argue against your use of anomalies, and for using the unadjusted data?

I disagree that the MAE is "far more pronounced pre-1910" in all the timeseries shown (c.f. the precipitation and the minimum air temperature). Can you be more quantitative? Is it necessary, or physically justified, to have a same time cut-off for all the timeseries?

Response 31: We recognise that the consistency over time may not be a result of the our interpolation methodology working well with minimal amounts of data, as we can see from the new Figure 3 that while the amount of data changes the spatial pattern in terms of the distances between stations has remained quite consistent, therefore we have removed this statement.

It is important to recognise the uncertainty layers provide information about the distribution of MAE across New Zealand; it is just that, in essence, the uncertainty weights the MAEs by distance to the data cells. This misunderstanding might explain your Comment 29 where you questioned the value of some example uncertainty maps, but we feel Figure 5b answers your query about the spatial distribution of error at a few times. This link between error and uncertainty was the basis of another comment, so please refer to Comment and Response 16 for how we have tried to clarify this, which we hope will make the relevance of the uncertainty examples more evident.

To us it makes logical sense that the interpolations between 1950-1980 that are within the timeframe of the climatology will have lower error. For example, consider two locations a and b where a is wetter during 1950-1980. Using climatologically aided interpolation we will always assume that a is wetter than b, but if by 2020 climate change means that is no longer the case, then the climatologically aided interpolation will begin to introduce errors.

You are correct that this is a limitation of climatologically aided interpolation, but we feel justified in our choice of method as it is consistent with the approaches of other researchers, and produces good results overall, even if you find them to be surprisingly consistent. Nevertheless, as we move further away from the time for which the climatology was constructed, we would expect our results to be worse, and this is what our results indicate. This is exactly why we have chosen to limit our data at 1910 such that we only extend 40 years either side of the climatology. The 1910 cut-off seems even more justifiable now that we have added data regarding data cell numbers and spatial arrangement (see Lines 190-193 and Figure 3) that clearly indicate that interpolations before 1910 will be suspect.

Comment 32: INTERPOLATION VALIDATION (lines 130-134): How does "the spatial variability of uncertainty in Figure 3" explain the variations in the "pattern of MAE over time" shown in Figure 4?

The statement that "while the number and location of weather stations may vary from month to month, the weather patterns may change dramatically making some months easier to interpolate than others" makes no meteorological or computational sense and seems highly speculative.

Response 32: Given our changes resulting from Comment and Response 29 this paragraph has become somewhat repetitive and redundant, so we have simply removed it to avoid confusion.

Comment 33: USING THE UNCERTAINTY DATA: This entire section is extremely weak. The choices to pull random values of monthly weather data from a uniform distribution and to have the range of that distribution equal the uncertainty, is not explained and is troubling since precipitation and temperature variations are not typically uniformly distributed. The claim that "the trend can be established precisely" is over-reach. The remainder of the section is no more convincing. This section would be best omitted, and interested users left to explore reasonable statistical uses on their own.

Response 33: Respectfully, we disagree. Our dataset is unique in providing uncertainty estimates in appropriate units if measurement for all interpolated values within the time-series. Therefore, given this is a novel type of data in this context, we feel some illustration of how this data can be used and visualised could be extremely helpful for those interested in making use of the data we have produced. We note that another reviewer thought this section was useful (please refer to Comment 5) so we have chosen to retain it, but we have added a note to stress that the approach we have used is only meant to be illustrative of the potential (see Lines 251-252).

Comment 34: LIMITATIONS AND FUTURE RECOMMENDATIONS (lines 180-182): I doubt that increasing the resolution to 100m is going to be an improvement if elevation is not accounted for.

Response 34: We have tried to clarify how elevation is accounted for indirectly through climatologically aided interpolation (please see Comment and Response 28), so hopefully that will help reduce some of the concerns about the inclusion of elevation. It seems self-evident to us that improving the resolution of underlying data, be it a climatology or a digital elevation model (DEM), will improve the resulting predictions, especially in mountainous regions where environmental conditions change over short distances. For example, when conducting an interpolation of

temperature based on elevation in mountains, a 100 m resolution DEM would surely perform better than a 1 km DEM as it will peel apart locations that are close together but at quite different elevations allowing them to have more localised predictions. The exact same logic applies with climatologically aided interpolation.

Comment 35: LIMITATIONS AND FUTURE RECOMMENDATIONS (lines 192-194): The statement about the possible dependence of MAE again raises the question of what benefits the climatological assistance is and why it outweighs the distortions it appears to generate.

Response 35: As we have already stated in Comment and Response 31, we feel justified in our choice of climatologically aided interpolation as it is consistent with the approaches of other researchers, it meets our requirements of working with minimal amounts of data, and produces good results overall. We do not suggest that this is the only or best method, and if someone can produce another open data set using another method that is superior to ours then we would very much welcome it! In the meantime, our data set clearly provides a significant step forward over options that are currently available.

Comment 36: DATA AVAILABILITY: This section needs a little more information than "non-proprietary file formats" are available. A sentence or two of explanation of what a *.7z file is and how to unpack it would be useful. Many if not most members of the meteorological and climatological communities are familiar with *.tif files as image files, and have no idea how to extract numerical data from them; that information here would be useful. Size estimates for the unpacked data would also be helpful.

Response 36: We are happy to provide more detailed information about file formats and sizes, but this is better done in the data archive at https://doi.org/10.7931/zmvz-xf30. Your comment about the geotiff file format is interesting, as we would consider it the most widely used open file format for geographical information, and hence is why it is used by other similar projects such as Chelsa and WorldClim listed in Table 1. Neverthless, we have added links to descriptions of both the .tif and .7z file formats for those people who are less familiar with these file formats and have added notes about the uncompressed data sizes too as requested.

Comment 37: REFERENCES: The references are difficult to read and would benefit from a blank line separating references. The format for datasets seems nonstandard, disagrees with the way they are cited in the text (c.f. GDAL/OGR, CliFlo). Information on how to obtain Leathwick et al. (2002) is needed.

Response 37: We have used the journal's Word template and EndNote style file, so the references are provided exactly as the journal requests them from us. We do take your point though, and will double-check this at the proofing stage. The full bibliographic information required by the journal citation format is provided for the Leathwick et al. (2002) book, so there is nothing else we can provide.

MINOR COMMENTS:

Comment 38: TABLE 1:

• The abstract of Fick and Hijmans (2017), the first data set listed in this table, says its spatial resolution is "approximately 1 km^2", not "≈4 km" as stated in the table.

• WorldClim 2 currently spans 1970-2000, not 1968-2018

(https://www.worldclim.org/data/worldclim21.html; Fick and Hijmans, 2017).

• "WorldClim 2" and "CHELSA" are improperly capitalized.

• The caption is misleading. Datasets such as WorldClim or CHELSA are not properly considered to be Year1–Year2 timeseries with monthly resolution, but rather sets (or "climatologies", to use the language of both references) of monthly averages spanning Year1–Year2. It would be helpful to indicate in the Table which of these datasets are in the form of long-term monthly means and which are long timeseries of monthly means like HOWZA.

Response 38: I suspect you may not be aware that WorldClim 2 and Chelsa provide not only the 30-year climatologies you refer to, but also a time-series of monthly weather grids (see: https://www.worldclim.org/data/monthlywth.html and https://chelsa-climate.org/timeseries/). So, the information in Table 1 is correct. Capitalisations have been corrected (see Table 1 and Line 25).

Comment 39: INTRO (lines 24 & 57, repeated throughout): The phrase "weather grids" is unusual in the earth sciences; "gridded weather data" would be more easily understood. Similarly, "spatial uncertainty grids" is unusual and confusing; "uncertainty at each gridpoint" or similar would be much clearer.

Response 39: We are happy with our use of terminology here, as a "grid" and "cell" as it is consistent with papers describing other datasets such as WorldClim 2 (Fick and Hijmans 2017), CHELSA (Karger et al. 2017), and TerraClimate (Abatzoglou et al. 2018). That said, to try to clarify the terms for those readers less familiar with them, we have extended our definitions (see Line 26).

Comment 40: INTRO (line 32-3): "In New Zealand" implies that the data sets are stored or created in New Zealand; "For New Zealand" would better indicate that the focus is data about New Zealand weather.

Response 40: Agreed, change made as suggested (Line 35).

Comment 41: INTRO (line 33): Whose "currently optimal criteria" specify 1km$^2$ and monthly resolution over the last century?

Response 41: We have reworded this sentence to try and explain that the dataset we are seeking to create has the optimum combination of all spatial and temporal characteristics available with all currently available open datasets (Lines 37-38).

Comment 42: INTRO (lines 52 & 57; repeated throughout manuscript): "Units of measurement" is unusual and open to confusion because some things are measured in one unit but typically reported in another. Referring to "the variables' units" is compact and clear, although their are many other ways to reword this.

Response 42: Good point; we have changed as suggested throughout.

Comment 43: INTERPOLATION VALIDATION (lines 121): "the number of data cells (which equates closely to the number of weather stations)" is a curiously vague statement. Please clarify.

Response 43: Apologies for the confusion here around data cells. To try to clarify this, we have extended the definition of data cells to make it clear that with the discrete from of natural neighbour interpolation weather stations data essentially become data cells, but where multiple stations occur

in a single cell, there is only one data cell that is the average of the weather station data (see Lines 128-131). Therefore, the number of weather stations and data cells is not necessarily the same, though at our 1-km grid resolution the two counts are almost identical as it is rare for two weather stations to be in such proximity, but it does happen and hence must be recognised and accounted for as we have done.

Comment 44: FIGURES 1 & 3: These would benefit from some visual separation of the top set of small multiples and the bottom set, e.g. a fine solid line splitting the horizontal white space between the top & bottom sets or a noticeably wider white space.

Response 44: Thanks for the suggestion. We have created some additional white space and have tried to further clarify the difference by labelling the distinct parts of the figure and have provided further explanation in the figure caption (see new Figures 2 and 5).

Comment 45: FIGURE 2: What do the small squares on all panels except 2b indicate? What are the units on 2d?

Response 45: The squares represent data cells, and we have added a clarification about that in the caption. As this is a hypothetical example for illustrative purposes there are no units, either for distance or the interpolation values, so we have added a note in the caption to clarify that too (Lines 478-479).

Comment 46: FIGURE 4: These would benefit from a small label on each panel (e.g. in the upper-right corner) indicating "rainfall", "air temperature", "minimum temperature", etc.

Response 46: Sure, we have added these to the panel labels – both in this figure and other figures that have the same form.

Comment 47: FIGURE 5: These 3 pairs of plots would benefit from a small label above or to the right of each giving a summary of the condition. Also, what is a "reliable trend" versus "no reliable trend"?

Response 47: We have added some labelling as requested and have reworded and expanded the caption to clarify what the plots show.

Comment 48: FIGURE 5: These would benefit from a simple subtitle or label in each part (a & b). Also, the meaning and utility of the inset colored grid to the west of New Zealand remains completely obscure to me. Finally, some indication of horizontal scale, position on the globe (lat/lon) and the words "New Zealand" in the caption would be useful.

Response 48: I am fairly sure you mean Figure 6 based on your comment, so we have added the information requested in the new Figure 9, along with a comment explaining the meaning of the two-dimensional value-uncertainty legend. We have also revised and expanded our description of the value-by-alpha cartographic method we have used here (Lines 238-240)

---

## Author Response (AR2)

8 June 2022

Dear Dr Gruber,

I would like to thank you and the reviewers once again for your time in helping us publish this dataset.

I have happily made the few technical typesetting and grammatical corrections that were identified by the most recent review.

I have also separated the Supplementary figures into a new supplemental file as was noted and requested during the previous review file validation.

Hopefully you will find the production files in good order, but if you require any further information or corrections please do let me know.

Kind regards,

Thomas Etherington